# Short-term neuronal and synaptic plasticity act in synergy for deviance detection in spiking networks

**Felix Benjamin Kern, Zenas C. Chao***

International Research Center for Neurointelligence (WPI-IRCN), The University of Tokyo, Tokyo, Japan

* zenas.c.chao@gmail.com

## Abstract

Sensory areas of cortex respond more strongly to infrequent stimuli when these violate previously established regularities, a phenomenon known as deviance detection (DD). Previous modeling work has mainly attempted to explain DD on the basis of synaptic plasticity. However, a large fraction of cortical neurons also exhibit firing rate adaptation, an underexplored potential mechanism. Here, we investigate DD in a spiking neuronal network model with two types of short-term plasticity, fast synaptic short-term depression (STD) and slower threshold adaptation (TA). We probe the model with an oddball stimulation paradigm and assess DD by evaluating the network responses. We find that TA is sufficient to elicit DD. It achieves this by habituating neurons near the stimulation site that respond earliest to the frequently presented standard stimulus (local fatigue), which diminishes the response and promotes the recovery (global fatigue) of the wider network. Further, we find a synergy effect between STD and TA, where they interact with each other to achieve greater DD than the sum of their individual effects. We show that this synergy is caused by the local fatigue added by STD, which inhibits the global response to the frequently presented stimulus, allowing greater recovery of TA-mediated global fatigue and making the network more responsive to the deviant stimulus. Finally, we show that the magnitude of DD strongly depends on the timescale of stimulation. We conclude that highly predictable information can be encoded in strong local fatigue, which allows greater global recovery and subsequent heightened sensitivity for DD.

**Data Availability Statement:** All program code and summary data necessary to reproduce this work is available at https://github.com/kernfel/plasticity-synergy-deviance-detection. In addition, the code

## Author summary

Our brains are constantly processing sensory information, but they must also be able to detect when something unexpected happens. This ability, known as deviance detection, is critical for filtering out irrelevant information and focusing on new or important stimuli. The precise mechanisms underpinning deviance detection remain unknown. While previous research and modeling studies have focused on synaptic changes as the mechanism for deviance detection, we hypothesized that firing rate adaptation, which is a non-synaptic plasticity mechanism that reduces a neuron's response to repeated stimulation, is also

is archived on Zenodo at https://doi.org/10.5281/zenodo.8333172.

**Funding:** FBK and ZCC were supported by World Premier International Research Center Initiative (WPI), MEXT, Japan (https://www.mext.go.jp/en/policy/science_technology/researchpromotion/title01/detail01/1374076.htm). The funders had no role in study design, data collection and analysis, decision to publish, or preparation of the manuscript.

**Competing interests:** The authors have no relevant financial or non-financial interests to disclose.

involved. Using spiking neural network simulations, we found that adaptation alone is sufficient to produce deviance detection. Additionally, we found that when neuronal adaptation was combined with synaptic short-term depression, the two mechanisms worked together to improve deviance detection. Our findings suggest that the brain may encode predictable information by rapidly adapting both neurons and synapses to frequent stimuli, allowing for heightened sensitivity to unexpected events. An improved understanding of how the brain detects unexpected events could lead to the development of new treatments for disorders such as autism, schizophrenia, or ADHD, which are associated with sensory processing difficulties.

## Introduction

Deviance detection (DD) is the ability of neural systems to identify statistically surprising sensory inputs, or violations of established regularities. It is a critical component of any signal processing cascade and is closely related to the concept of prediction errors in the predictive processing framework [1–4], which has been suggested to underlie most of cognition [5, 6]. DD has been studied widely across sensory systems including the auditory [3, 7, 8], visual [9, 10], somatosensory [11] domains and beyond. Impairments in DD have been reported in various disorders [4], including autism [12–16], schizophrenia [17–20], ADHD [21–23] and others, highlighting the need to better understand its mechanistic underpinnings.

Studies of DD must carefully distinguish between stimulus-specific adaptation (SSA) and true deviance detection [24]. The former refers to the reduction of the response to a frequently presented stimulus and can be explained by any adaptive mechanism [25, 26]. The latter refers to the observation of a larger response to an infrequent stimulus when it is statistically surprising, e.g., due to interrupting a series of regular stimuli, compared to an unsurprising control condition. Here, we study the latter, true DD, which has been suggested to rely on a comparison between prediction signals and received inputs [3, 24, 27]. In the remainder of this paper, the term "true DD" [7, 24, 28] is used to refer to carefully controlled results, where stimulus-specific adaptation is excluded by experimental design. By contrast, unmarked "DD" is used to refer to the broader concept of deviance detection in the sense of a computational goal, which can include both SSA and true DD.

Existing computational models of DD place their primary focus on synaptic plasticity as the causal mechanism, acting either along feed-forward pathways to reduce transmission, or in recurrent connections to dampen the lateral propagation of activity. While models using long-term plasticity have been proposed [29, 30], the rapid establishment of DD in vivo [31] suggests that fast-acting short-term plasticity mechanisms play a key role. Several models using short-term synaptic plasticity have been proposed [32–34], finding good agreement with physiological data. However, short-term plasticity also occurs in neuronal excitability, with many neurons exhibiting an experience-dependent transient reduction in intrinsic neuronal sensitivity independent of synaptic function [35–37]. Despite operating at similar time scales to short-term synaptic plasticity, the effects of intrinsic neuronal plasticity on DD is unexplored. Previous modeling work has only used neuronal plasticity as a side effect of the chosen neuron model [32] or rejected it a priori as not sufficiently stimulus specific, and therefore incapable of producing either SSA or true DD [34]. One exception is the work by [26], which showed that neuronal plasticity in a multi-layer firing rate model could give rise to greater responses to infrequent, surprising stimuli, but did not distinguish between true DD and SSA.

Furthermore, it remains unknown how short-term synaptic and neuronal plasticity, which modulate the input to and the output from a neuron, respectively, work together in a network for DD.

Neural systems that support true DD include mammalian cortical and subcortical brain areas [3, 27, 38]. Perhaps the simplest system known to exhibit true DD, however, is neuronal cell culture, which has recently been probed with the established paradigm for true DD [39]. Here, we take inspiration from the structural simplicity of this system, and approach true DD using a network model of spiking neurons with threshold adaptation and short-term synaptic depression. We pursue two aims: Firstly, we aim to show that neuronal adaptation can induce true DD in much the same way as synaptic depression and investigate the mechanistic basis of this effect in detail. Secondly, we aim to investigate the interactions between neuronal and synaptic short-term plasticity in the context of DD. We show that the two mechanisms work similarly and can synergize to enhance their effect. Our results widen the scope of possible mechanisms of DD and highlight the need to consider interactions between various forms of plasticity.

## Methods

### Model

We modeled neurons as leaky integrate-and-fire units whose membrane potential $V_m$ at time $t$ followed

$$\tau_m \frac{dV_m}{dt} = (V_{rest} - V_m) + R_m I_{syn}(t) \tag{1}$$

with membrane time constant $\tau_m$ = 30 ms, membrane resistance $R_m$ = 100 MΩ, and resting membrane potential $V_{rest}$ = −60 mV. Synaptic currents were modeled in a conductance-based manner, following

$$I_{syn} = g_e(E_e - V_m) + g_i(E_i - V_m) \tag{2}$$

with excitatory reversal potential $E_e$ = 0 mV and inhibitory reversal potential $E_i$ = −100 mV. Synaptic conductances evolved according to

$$\tau_e \frac{dg_e}{dt} = -g_e + \sum_{j \in E} U x_j w \delta(t - \hat{t}_j)$$
$$\tau_i \frac{dg_i}{dt} = -g_i + \sum_{j \in I} w \delta(t - \hat{t}_j) \tag{3}$$

with excitatory time constant $\tau_e$ = 2 ms, echoing AMPA receptor dynamics [40], inhibitory time constant $\tau_i$ = 4 ms, echoing GABA-A receptor dynamics [41], the sets of presynaptic excitatory and inhibitory neurons $E$ and $I$, respectively, synaptic weight $w$, the Dirac delta function $\delta(\cdot)$, and spike times $\hat{t}$.

Excitatory, but not inhibitory, synapses were subject to short-term depression (STD), simplified from [42]; since synaptic transmission was not stochastic, we modeled the depression variable $x_j$ as a property of the presynaptic neuron $j$,

$$\tau_x \frac{dx_j}{dt} = (1 - x_j) - U x_j \delta(t - \hat{t}) \tag{4}$$

with recovery time constant $\tau_x$ = 150 ms, consistent with synaptic depression in auditory cortex [43, 44], and release fraction $U$ = 0.4.

 

When a neuron's membrane potential reached the firing threshold $V_\theta$, a spike was emitted, and the potential was clamped to $V_m = V_{reset} = -74$ mV for a fixed refractory period of 3 ms (excitatory neurons). Inhibitory neurons were modeled as fast-spiking cells with a refractory period of 2 ms and a constant firing threshold $V_\theta = \theta_0 = -54$ mV [45]. In contrast, excitatory neurons were modeled with threshold adaptation (TA) as in [46], following

$$
\begin{aligned}
V_\theta &= \theta_0 + \theta(t) \\
\tau_\theta \frac{d\theta}{dt} &= -\theta + \hat{\theta}\delta(t - \hat{t})
\end{aligned}
\tag{5}
$$

with increment $\hat{\theta} = 1$ mV and decay time constant $\tau_\theta = 1$ s [47–50].

Fig 1A shows the internal and presynaptic dynamics of an excitatory neuron in a contrived example for the purpose of illustration. The model was implemented in Brian2 [51], accelerated with Brian2GeNN [52], and simulated with an integration time step of 1 ms. Synaptic delay was not modeled explicitly, but spikes were delivered, and voltages reset, in the time step following spike emission. We simulated neurons without any stochasticity in either their inputs or their parameters, other than network structure (described below), in order to ensure

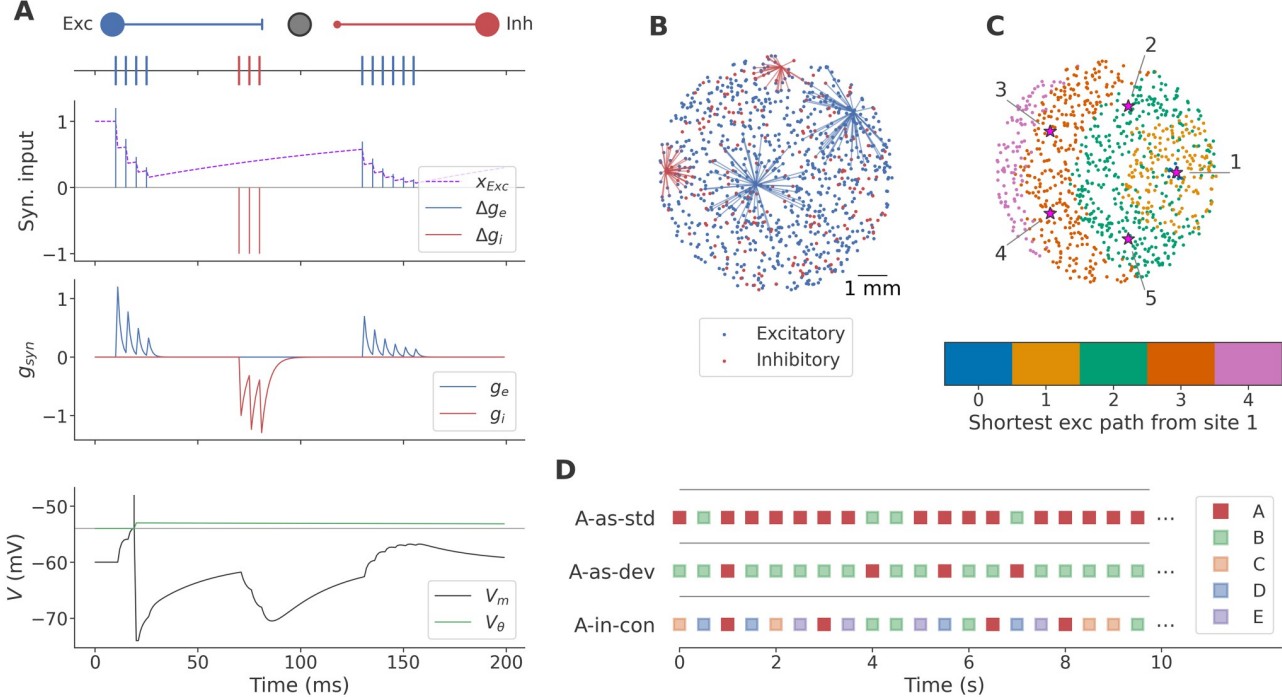

**Fig 1. Model and paradigm. A**: Membrane and synaptic dynamics of a neuron innervated by an excitatory and an inhibitory neuron firing predetermined spikes, illustrated above the plots. The excitatory connection has weight $w = 0.3$ nS for demonstration purposes only; a single connection with $w = 0.1$ nS is normally unable to evoke postsynaptic firing. Top: Inputs to the (post-)synaptic conductances $g_e$ and $g_i$ (vertical bars), and the STD depression variable $x_{Exc}$ of the excitatory presynaptic neuron (dashed line). Note that excitatory inputs are equal to $Ux_{Exc}w$ as per Eq (3). Middle: Excitatory and inhibitory synaptic conductances ($g_e$ and $g_i$, respectively). Bottom: Membrane voltage, firing threshold baseline (gray) and adaptive threshold, demonstrating TA in the excitatory postsynaptic neuron. **B**: Sample network layout, showing the spatial location of excitatory and inhibitory neurons, and all outgoing synaptic connections from two neurons of each type. **C**: Distance from stimulation site 1 in terms of minimum number of excitatory synapses. Stimulation is provided to the 10 neurons nearest to the center of each of 5 stimulation sites (large circles), which are regularly spaced on a circle with a radius of 2.5 mm. **D**: Illustration of the stimulation paradigm. Each marker represents a stimulation, with the stimulus identity indicated by color. The target stimulus (A) is represented as solid red, and non-target stimuli are represented with faded colors. Three randomized sequences were presented, containing 80% A and 20% B (*A-as-std*), 20% A and 80% B (*A-as-dev*), and 20% of each of 5 stimuli, including A and B (*A-in-con*). Note that, while the onset asynchrony of 500 ms is represented faithfully, the horizontal extent of the stimulation markers has no meaning; stimuli were presented instantaneously rather than over an extended period.

 

that our results were driven purely by the experimental paradigm, rather than any model-internal sources of noise.

800 excitatory and 200 inhibitory neurons were placed randomly in a two-dimensional circular space with a radius of 4 mm in rough analogy to neuronal cell culture. Neurons formed synapses of weight $w$ = 0.1 nS with 50 randomly selected postsynaptic partners within a range of 2 mm (excitatory) or 1 mm (inhibitory), reflecting the notion that excitatory neurons project over longer distances, while inhibitory neurons mainly connect to their local neighborhood. The particular values were chosen to achieve reliably large stimulus responses without giving rise to instabilities or ongoing activity. Connectivity and weights remained fixed throughout. Stimulation sites were evenly distributed 2.5 mm from the center of the dish, and stimulation delivered a one-time increase in $g_e$ to the 10 closest (inhibitory or excitatory) neurons, sufficient to trigger 2–3 spikes each. Fig 1B shows a representative example of the resulting network structure. While intentionally not an exact reproduction of the situation in cell culture—we aim to model a generic phenomenon, not a particular system—the network architecture and stimulus delivery paradigm crucially omit certain features (e.g., structural complexity, tuning curves, or adaptation in the input stimulus) that appear to be unnecessary to achieve true DD. In Fig 1C, we show the distance from the neurons stimulated at site 1 to each neuron in this network, in terms of the minimum number of excitatory synapses that need to be traversed to reach the target, in order to give an intuition for how activity may spread. Networks were pseudo-randomly generated according to the above scheme and screened for minimal stimulus response: Networks where fewer than 500 neurons responded to stimulation at any of the five sites after full recovery were discarded. In total, 30 networks were used for the experiments described below.

## Paradigm

To investigate true DD, we used a classical oddball paradigm as illustrated in Fig 1D, presenting a total of 500 stimuli at regular intervals of 500 ms. Target stimuli (labeled "A" or "target" hereafter) and distractor stimuli (labeled "B" to "E") were presented in three randomized sequences: The "standard" sequence (*A-as-std*, abbreviated to *A:std* where appropriate), consisting of 400 presentations of A and 100 presentations of B; the "deviant" sequence (*A-as-dev* or *A:dev*) with 100 A and 400 B, and the many-standards control sequence (*A-in-con* or *A:con*) with 100 presentations of each of the stimuli A through E. In this way, A is presented as the predominant and therefore expected stimulus (*A-as-std*), as an infrequent violation of an expectation (of B, *A-as-dev*), and, to control for pure adaptation effects, as an equally infrequent stimulus with no strong expectation of any particular input (*A-in-con*).

In each network, we ran 4 oddball sequences, using the following stimulus site pairings as target/non-target (A/B) pairs: 1/2, 2/1, 3/5, and 5/3, yielding both *A-as-dev* and *A-as-std* sequences for all 4 stimulus sites involved. In addition, we ran one *A-in-con* sequence involving all stimulus sites, the results of which were shared between the data sets from one network. Networks were reset to a fully recovered state between sequence presentations to guarantee their independence. This yielded 4 complete data sets per network, or a total of 120 data sets across all networks.

In order to disentangle the effects of STD and TA, we ran all simulations in four conditions: Without any short-term plasticity, with either TA only or STD only, and with both STD and TA (labeled STD+TA in the following, and corresponding to the full model as described above). To turn off STD, we replaced Eq (4) with a constant $x$ = 1, but retained the scaling of inputs to $g_e$ with $U$ in Eq (3). This corresponds to immediate recovery ($\tau_x$ = 0) and was done to maintain the magnitude of the excitatory postsynaptic potential (EPSP) in the recovered state

regardless of whether STD was turned on or off. To turn off TA, we replaced Eq (5) with a constant $V_\theta = \theta_0$, making the firing threshold completely unadaptive.

## Sample network

We present most of our analysis at two levels: We first detail the processes leading to true DD on a sample network, then confirm the highlighted trends statistically with reference to the full 120 data sets. The sample network was chosen to be roughly representative of the analyzed data sets as follows: In each data set, we assessed the average number of spikes fired in response to stimulation with A and B in the *A-in-con* sequence. We considered only data sets whose response size for both A and B was within one standard deviation of the mean response size across data sets, then hand-picked a sample data set (a network and stimulus assignment, hereafter referred to as the "sample network") that exhibited both an larger deviance detection index (see Eq (12) below) in STD+TA over TA only, and greater *A-in-con* than *A-as-std* responses in both STD+TA and TA only conditions.

## Notation and quantities

**Notation.** Throughout this paper, we refer to three main quantities of interest: the stimulus response magnitude **R**, the threshold adaptation value **T**, and the short-term depression value **D**. We define the response **R** as the mean number of spikes fired across all indicated trials (see below), i.e., the average trial response. We define **T** as the mean across trials of $\theta(t = 0)$, i.e., the average TA voltage at the beginning of trials, immediately before stimulation. Lastly, the related STD quantity **D** is introduced in Eq (11) below.

We employ sub- and superscripts to indicate the precise quantity as follows, using **R** for illustration: ${}^{seq}\boldsymbol{R}_p^S$ refers to the response of neuron(s) $p$, averaged across all trials of type $S$ in the sequence *seq*. We use $p$ to either identify individual neurons (e.g., $i$ or $j$), or a subpopulation (e.g., *exc*) or the whole network (no index), in which case the designated quantity is the median of the individual quantities ${}^{seq}\boldsymbol{R}_i^S$ across the indicated set of neurons. The trial type $S$ is either $A$ or $B$, referring to trials with stimulation at these sites, or $N$, indicating all *n*on-target trials B through E. Finally, we use the notation Δ**R** to refer to the sequence contrast ${}^{A\text{-}as\text{-}dev}\boldsymbol{R} - {}^{A\text{-}in\text{-}con}\boldsymbol{R}$, and $\overline{\boldsymbol{R}_p}$ to indicate the mean, rather than median, of the individual quantities in a population $p$.

**Contribution of TA to greater activity.** To quantify the contribution of lower thresholds to greater activity in *A-as-dev* compared to *A-in-con*, we first calculated the spike probability $\overline{P}$, average membrane voltage $\overline{V_m}$, and average threshold adaptation $\overline{\theta}$, across target trials for each neuron $i$, time step $t$, and sequence seq:

$$
\begin{aligned}
\overline{P}(i, t, \text{seq}) &= \frac{1}{N_{\text{trials}}} \sum_{j \in \text{trials}} \mathbb{1}(i, t, \text{seq}, j) \\
\overline{V_m}(i, t, \text{seq}) &= \frac{1}{N_{\text{trials}}} \sum_{j \in \text{trials}} V_m(i, t, \text{seq}, j) \\
\overline{\theta}(i, t, \text{seq}) &= \frac{1}{N_{\text{trials}}} \sum_{j \in \text{trials}} \theta(i, t, \text{seq}, j)
\end{aligned}
\tag{6}
$$

where

$$
\mathbb{1}(i, t, \text{seq}, j) = \begin{cases} 1 & \text{if neuron } i \text{ fired at time } t \text{ in trial } j \text{ of sequence seq} \\ 0 & \text{otherwise} \end{cases}
$$

Then, we contrasted these values between *A-as-dev* and *A-in-con*:

$$
\begin{aligned}
\Delta P(i,t) &= \overline{P}(i,t,A\text{-}as\text{-}dev) - \overline{P}(i,t,A\text{-}in\text{-}con) \\
\Delta \theta(i,t) &= \overline{\theta}(i,t,A\text{-}as\text{-}dev) - \overline{\theta}(i,t,A\text{-}in\text{-}con) \\
\Delta V_m(i,t) &= \overline{V_m}(i,t,A\text{-}as\text{-}dev) - \overline{V_m}(i,t,A\text{-}in\text{-}con)
\end{aligned}
\tag{7}
$$

Next, we focused on bins (neurons and time steps) with greater firing in *A-as-dev*, i.e., $\Delta P(i,t) > 0$, and asked to what extent differences in $V_m$ and $\theta$ contributed to this difference in firing. Noticing that only lower thresholds ($\Delta\theta < 0$) and higher voltages ($\Delta V_m > 0$) can cause greater activity, we ignored values outside of these limits, constructing contribution scores

$$
\begin{aligned}
C_{V_m}(i,t) &= \frac{[\Delta V_m(i,t)]^+}{[-\Delta\theta(i,t)]^+ + [\Delta V_m(i,t)]^+} \\
C_\theta(i,t) = 1 - C_{V_m} &= \frac{[-\Delta\theta(i,t)]^+}{[-\Delta\theta(i,t)]^+ + [\Delta V_m(i,t)]^+}
\end{aligned}
\tag{8}
$$

where

$$
[x]^+ = \begin{cases} x & \text{if } x > 0 \\ 0 & \text{otherwise} \end{cases}
$$

Finally, to control for the difference in spike probability, we weighted the contributions by $\Delta P$, following the intuition that greater differences in activity are of greater interest, and averaged across neurons, yielding time courses $\overline{C_{V_m}}$ and $\overline{C_\theta}$ describing the network-wide contribution of voltage and threshold to the deviant response:

$$
\begin{aligned}
\overline{C_{V_m}}(t) &= \frac{1}{|M(t)|} \sum_{i \in M(t)} C_{V_m}(i,t)\Delta P(i,t) \\
\overline{C_\theta}(t) &= \frac{1}{|M(t)|} \sum_{i \in M(t)} C_\theta(i,t)\Delta P(i,t)
\end{aligned}
\tag{9}
$$

where the mask $M$ selects the bins of interest, i.e., $M(t) = \{i \in \text{exc. neurons} \mid \Delta P(i,t) > 0\}$.

**Quantifying the effect of STD.** To quantify the effect of STD on the activity of a given neuron (or, alternatively, on its membrane potential $V_m$) in analogy to the effect of TA captured in $T$, we devised a measure $D$ to estimate the amount of postsynaptic membrane potential lost due to presynaptic depression, mirroring $T$ in the sense that decreasing $V_m$ and increasing the firing threshold both increase the depolarization necessary to make a neuron fire. A measure of STD is necessarily activity-dependent—a postsynaptic neuron's membrane potential is driven only by presynaptic partners that fire—so it must be calculated based on observed activity. In order to respect this notion while at the same time minimizing the influence of activity differences between sequences, we used the average response

$$
\widehat{R}_j^S = \frac{1}{3}\left( {}^{A\text{-}as\text{-}dev}R_j^S + {}^{A\text{-}in\text{-}con}R_j^S + {}^{A\text{-}as\text{-}std}R_j^S \right)
\tag{10}
$$

of a neuron $j$ to a given stimulus $S$ as a measure of the presynaptic drive. Before depression, a single excitatory presynaptic spike evokes a postsynaptic potential (EPSP) peaking at approximately $k = 1.4$ mV at $V_m = V_{rest}$ (numerical solution; recall that $w = 0.1$ nS for all synapses), and STD linearly scales this EPSP by presynaptic $x_j$. Thus, we estimated the average peak EPSP evoked by a spike in presynaptic neuron $j$ as $\widehat{R}_j^S x_j^S k$, where we ignored added depression due to

repeated spikes within a trial and instead defined $x_j^s$ as the mean depression at the start of trials. Finally, summing over presynaptic neurons, we estimated the peak EPSP lost to depression in each neuron $i$ as

$$\boldsymbol{D}_i^s = \sum_{j \in pre} \widehat{\boldsymbol{R}}_j^s \left(1 - x_j^s\right) k \tag{11}$$

Notice that $\boldsymbol{D}$ is measured in volts due to scaling with $k$, and is therefore roughly comparable to $\boldsymbol{T}$, albeit more approximate.

**Quantifying true deviance detection.**   Finally, we define true DD as a network's ability to pick out unexpected inputs from an otherwise homogeneous or otherwise unsurprising stream of inputs. Following related research [39, 53, 54], we were careful to exclude effects due to stimulus identity (i.e., avoiding direct comparisons between different stimuli) or due to adaptation to frequent stimulus presentation (i.e., avoiding direct comparisons of A in the *A-as-dev* sequence to A in the *A-as-std* sequence). In other words, we used the *A-in-con* sequence as a neutral baseline where the target stimulus was presented no more often than in the *A-as-dev* sequence, but no expectation of a competing stimulus B could be formed. To quantify true DD, we compared the mean number of spikes fired per neuron in response to target stimulus A in the *A-as-dev* and *A-in-con* sequences, denoted as $\overline{^{A-as-dev}\boldsymbol{R}^A}$ and $\overline{^{A-in-con}\boldsymbol{R}^A}$, respectively, and define a deviance detection index (DDI) as

$$DDI = \frac{\overline{^{A-as-dev}\boldsymbol{R}^A} - \overline{^{A-in-con}\boldsymbol{R}^A}}{\overline{^{A-as-dev}\boldsymbol{R}^A} + \overline{^{A-in-con}\boldsymbol{R}^A}} \tag{12}$$

## Results

### Deviance detection

To show that our model is capable of true DD, we first show the responses of the sample network in all sequences (*A-as-dev*, *A-in-con*, and *A-as-std*) in Fig 2A. Here, each row corresponds to one trial, aligned to stimulus input at t = 0, with the number of spikes in each millisecond bin indicated by color. All responses share an early activity pattern, which derives from the stimulated neurons and their immediate postsynaptic targets, but differences soon start to emerge. The *A-as-dev* sequence responses, shown at the top of the plot, exhibit a clear peak in activity around 15 ms after stimulation. The *A-as-std* sequence responses show a later, more diffuse activity peak around 20 ms, and the *A-in-con* sequence responses lack any obvious activity cluster. We sum these patterns up in Fig 2B, where we show the mean network activity, in terms of spikes per millisecond bin, across trials of each type. Clearly, the target response in the *A-as-dev* sequence is greater than either the *A-in-con* or the *A-as-std* response.

Individual neurons responded in a similar pattern, shown by way of the distribution of single-neuron response magnitudes in Fig 2C. The *A-as-dev* sequence targets evoked higher responses in a large portion of the network. Subtracting the *A-as-dev* response from either *A-in-con* or *A-as-std*, however, we see that a bimodal distribution emerges, with some neurons showing little to no difference in their response. This second, unchanging portion consists mainly of neurons that respond very little in any sequence. Removing such unresponsive neurons (specifically, the 204 neurons that fire less than 0.1 spikes per trial in any sequence), this portion largely disappears, and the distributions become unimodal and largely positive, indicating that neurons that are sensitive to the target stimulus respond most strongly in the *A-as-dev* sequence.

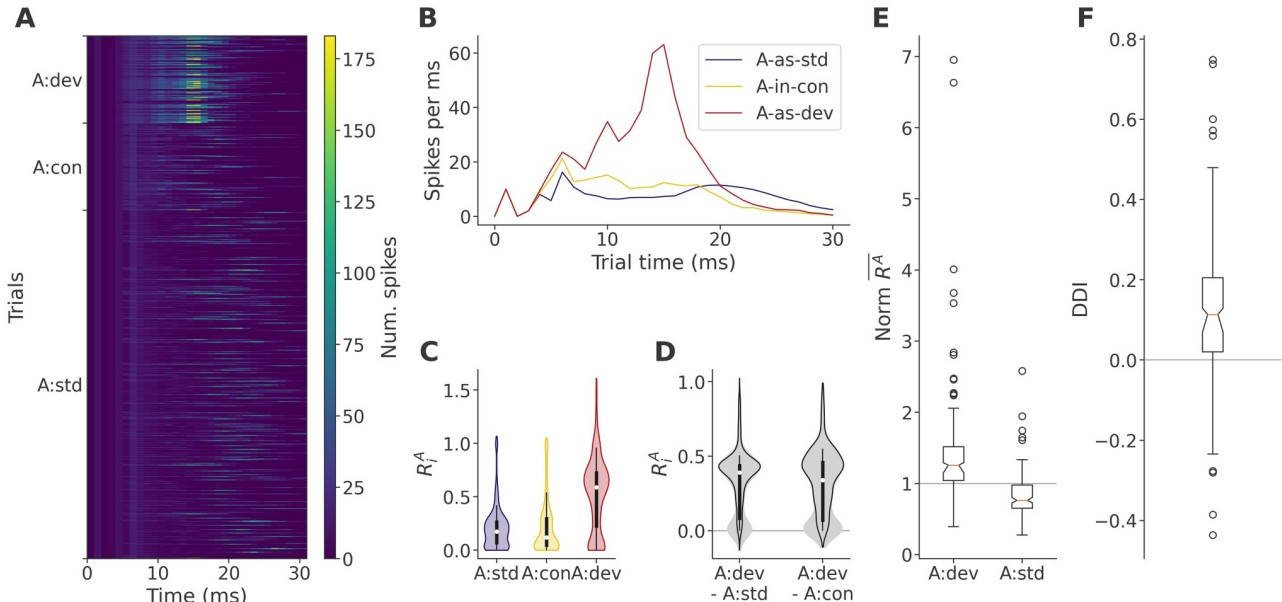

**Fig 2. Deviance detection in our model. A**: Sample network target response intensity by trial type. Each horizontal line of the plot summarizes the network response in one target trial in terms of spikes per integration time step. Target trials are taken from the *A-as-std* (400), *A-in-con* (100), and *A-as-dev* (100) sequences, and sorted by sequence time (earliest trials are at the bottom of each section). **B**: Response summary, showing average network-wide spike counts in target trials in each sequence. The underlying data are the same as in panel A. Note how the *A-as-dev* response is very large, followed by an intermediate *A-in-con* response and a low *A-as-std* response. **C**: Distribution of single neuron responses (in mean spikes per trial) in all sequences. The 10 stimulated neurons responded more strongly and are omitted. The inset bars represent quartiles (thick bars) and first and last deciles (thin bars), and the median is indicated with a white dot. **D**: Distribution of single neuron response differences between *A-as-dev* and *A-as-std*, *A-in-con* sequences (all neurons, grey, no outline, no quantiles). The distribution shown as a black outline, which is also represented with quantiles as in panel C, excludes neurons that fire less than 0.1 spikes per trial in any sequence. **E**: Mean response sizes $\overline{^{A\text{-}as\text{-}dev}R^A}$ and $\overline{^{A\text{-}as\text{-}std}R^A}$, normalized by $\overline{^{A\text{-}in\text{-}con}R^A}$, across data sets ($n = 120$). In this and later boxplots, the median is indicated with an orange line, notches indicate the 95% confidence interval for the median calculated by bootstrap with 10000 iterations, boxes indicate the inter-quartile range (IQR), whiskers extend to 1.5 times the IQR or to the data extrema, whichever is less, and fliers indicate individual data points beyond the whisker end points. **F**: Deviance detection indices (DDI, see Eq (12)) across data sets.

We could readily identify similar response patterns emerging across most data sets, though details of the temporal response profile varied widely. To show that true DD is robust to network and stimulus identity, we calculated the average spike count per neuron per target trial for each network and stimulus assignment, and compared these across sequences. Fig 2E shows a summary of the average *A-as-dev* and *A-as-std* response sizes, normalized by the response in the *A-in-con* sequence. The data clearly confirmed that *A-as-dev* responses were larger than *A-in-con* ($Z = 7.21$, $p = 2.75 \times 10^{-13}$; one-sided Wilcoxon signed-rank test, $n = 120$), and *A-as-std* responses smaller ($Z = -7.71$, $p = 6.42 \times 10^{-15}$). Accordingly, the DDI (Fig 2F) was positive in most networks ($Z = 6.12$, $p = 4.75 \times 10^{-10}$, median 0.114). This shows that our model was well behaved and suitable for an investigation of true DD.

Having confirmed the presence of true DD in our model, we turned to an ablation approach to try to identify how the two short-term plasticity mechanisms in the model, STD and TA, enabled the networks to learn relevant regularities. Fig 3A shows the responses to target stimulation of the sample network under ablation. With no plasticity, the responses are indistinguishable between sequences, yielding a DDI of 0, and indicating that stimulation did not have any long-lasting effects in membrane voltage or synaptic conductance. With STD only, the response in the *A-as-dev* sequence developed earlier and was very slightly larger, yielding a DDI of 0.03. With TA only, the response became noticeably smaller, but the sequences became readily distinguishable, with high *A-as-dev*, intermediate *A-in-con*, and low

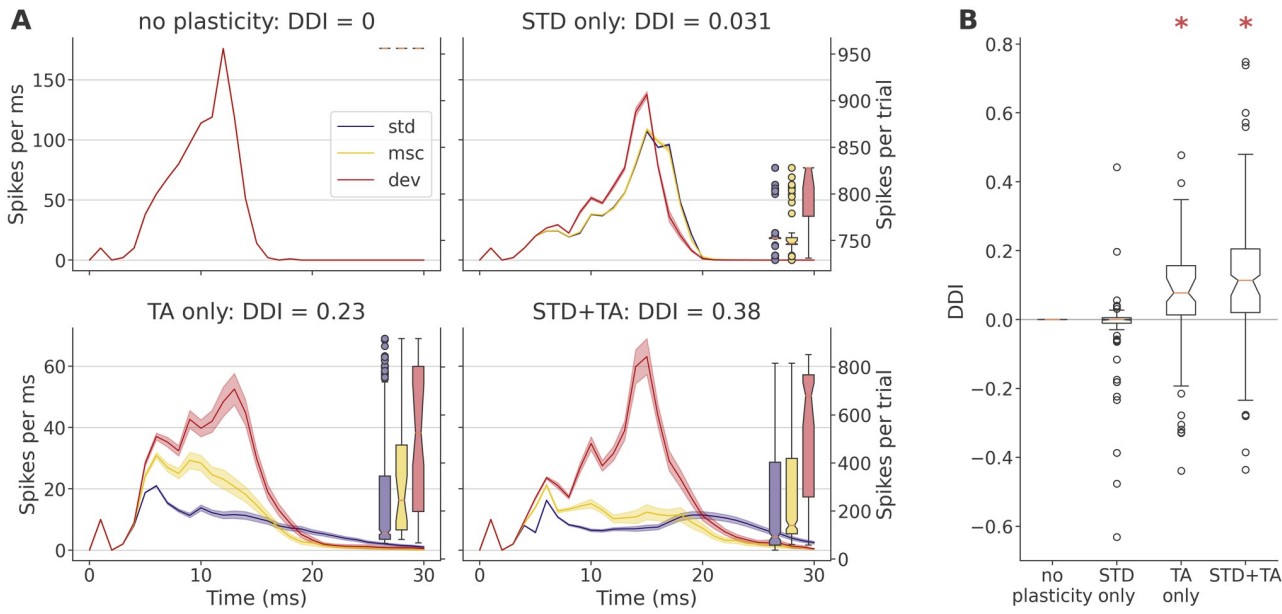

**Fig 3. Deviance detection under model ablation. A**: Sample network responses to target stimulation, with the deviance detection index in each instance noted in the plot titles. Lines show the average (trial mean +- SEM, left axis) network-wide spike counts in target trials in each sequence. Boxes summarize the spike count totals per trial (right axis), the means of which form the basis for index calculations (see panel B). Note the different axis scaling on the top row (without TA) and the bottom row (with TA). **B**: Deviance detection indices across data sets. Asterisks indicate data greater than 0, established with a one-tailed Wilcoxon signed-rank test at a significance level 0.05. See main text for detailed statistics.

*A-as-std* responses, yielding a DDI of 0.19. Finally, in the full model, the *A-as-dev* response stood out more clearly, leaving much smaller *A-in-con* and *A-as-std* responses and yielding a DDI of 0.4.

Finally, across data sets, the DDI (Fig 3B) was exactly 0 as expected in the ablation with no plasticity, not significantly greater than 0 with STD only ($Z = -1.12$, $p = 0.869$, median $-2.1 \times 10^{-4}$; one-sided Wilcoxon, $n = 120$), but clearly positive for TA only ($Z = 5.81$, $p = 3.1 \times 10^{-9}$, median 0.077) and the full model ($Z = 6.12$, $p = 4.75 \times 10^{-10}$, median 0.114). To our surprise, the DDI of the full model was significantly greater than that of either plasticity mechanism alone (STD: $Z = 6.47$, $p = 4.95 \times 10^{-11}$; TA: $Z = 3.24$, $p = 6.04 \times 10^{-4}$), and greater too than the sum of the DDI values across both STD only and TA only models ($Z = 3.92$, $p = 4.52 \times 10^{-5}$). This suggests that TA and STD act not as independent mechanisms in this paradigm, but rather interact constructively to enhance the network's ability to encode input regularities.

In the following, we will first build an understanding of how TA leads to true DD when acting alone, i.e., in the ablated TA only model, working backwards from the greater target response in *A-as-dev*, identifying and tracing causal factors step by step. Then, we will build on this understanding to elucidate how the addition of STD, which was shown above to be ineffective for true DD on its own, can lead to an increased deviant response in the full model.

## The role of TA in true deviance detection

To understand why the same stimulus A evoked a higher response in *A-as-dev* than in *A-in-con* with TA only, we first compared the average responses in the two sequences, sorting neurons by their response onset to permit a high-resolution view of the stimulus response as it propagates through the network. As shown in Fig 4A (left and middle), we found that the

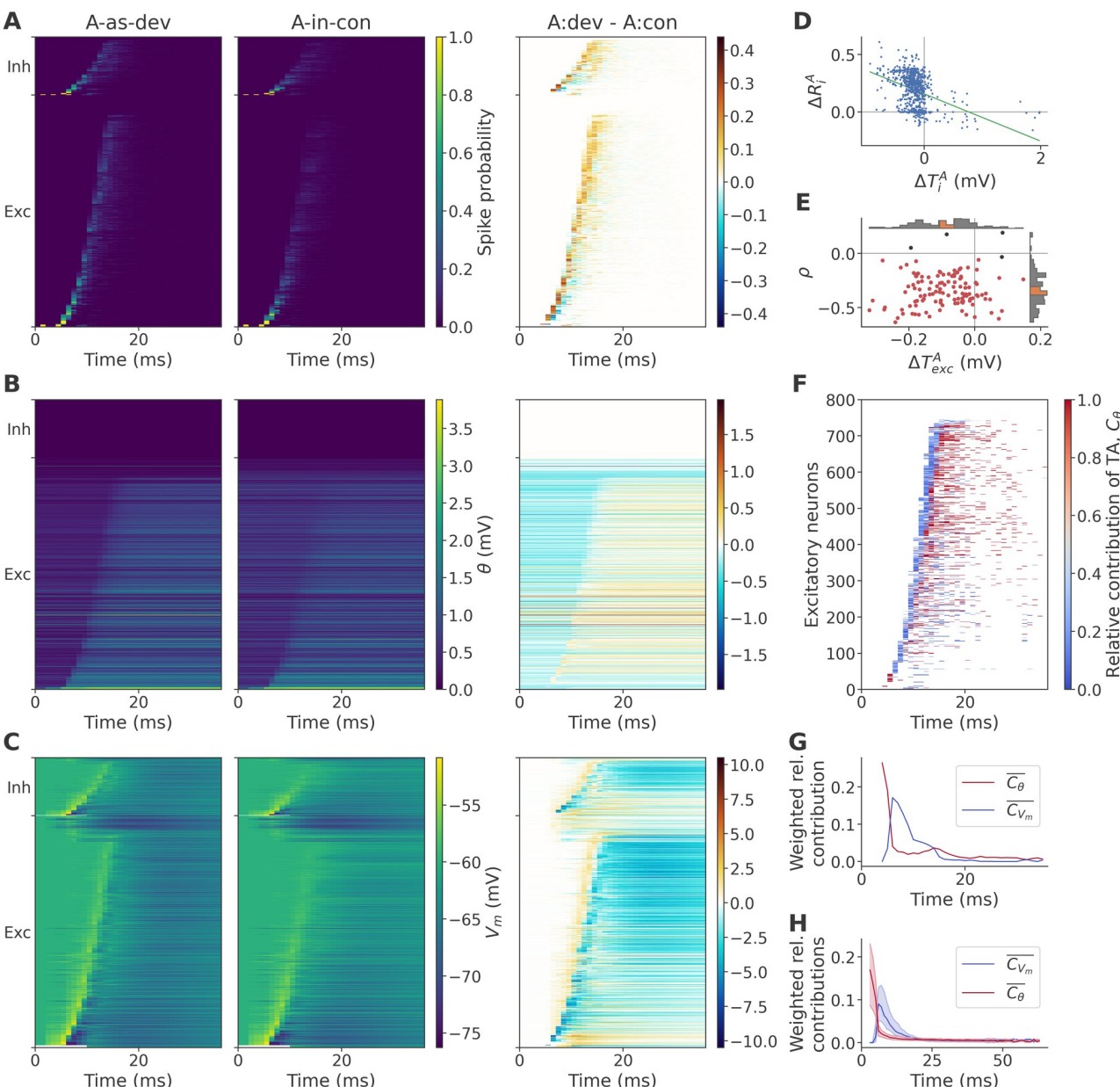

**Fig 4. Responses to *A-as-dev* target trials are larger due to lower initial θ. A**: Post-stimulus spike histogram across target trials in the sample network, showing trial time along the horizontal axis, and neurons along the vertical, sorted by the time of the first recorded spike across all trials and sequences. Left and middle column, target trials in the *A-as-dev* sequence and *A-in-con* sequence, respectively (see Eq (6)); right column, contrast between these two (Eq (7)). **B**: Target trial average of the threshold adaptation voltage θ, with neuron order and columns as in panel A. **C**: Target trial average of the membrane potential $V_m$, with neuron order and columns as in panel A. The first 10 ms of the neurons under direct stimulation are masked out to avoid saturation of the colormap. **D**: Relationship across excitatory neurons of the sample network between the contrast (*A-as-dev—A-in-con*) in θ at the start of target trials ($\Delta T_i^A$) and the contrast in the average number of spikes fired in target trials ($\Delta R_i^A$), and associated linear regression. **E**: Pearson correlation coefficients ρ of the relationship shown in the previous panel ($\Delta T_i^A$ vs $\Delta R_i^A$ across excitatory neurons), plotted against the corresponding $\Delta T_{exc}^A$ across data sets. Each marker corresponds to one network and target stimulus. Red markers indicate a significant negative correlation (one-sided Wald test, significance level 0.05). Inset histograms show the distribution of the corresponding values, with orange portions indicating the 95% confidence interval of the median, calculated by bootstrap with 10000 iterations. **F**: Contribution $C_θ$ of TA to greater firing in *A-as-dev* than *A-as-std*, see Eq (8). TA contributions that exceed the contribution of membrane voltage are shown in red. **G**: Contributions of TA (red) and $V_m$ (blue), weighted by difference in spike probability, and averaged across neurons ($\overline{C}$, see Eq (9)). **H**: Weighted average contributions $\overline{C}$ of TA and $V_m$ as in panel G, summarized across data sets. Plotted are the median (solid line) and inter-quartile range (shaded area).

response developed in a qualitatively similar fashion in both sequences. The contrast (right) clearly revealed higher *A-as-dev* activity, particularly in the later parts of the response, as shown by the predominance of orange areas. Since the stimulus was identical in both sequences, and TA was the only plasticity mechanism in play, these differences had to be due to (1) different firing thresholds, and/or (2) different synaptic currents as a consequence of different prior activity within the trial.

We quantified the former with the strength of adaptation $\theta$ (Eq (5)), i.e., the amount by which the firing threshold was raised due to TA. As shown in Fig 4B, thresholds at the start of target *A-as-dev* trials were generally clearly lower relative to *A-in-con*, visible both in the raw data (left and middle panels) and in the contrast (right panel, early blue portions). On the other hand, once the bulk of activity had occurred, many neurons showed higher thresholds in *A-as-dev* (right panel, later orange portions), since they spiked more often. Finally, note that inhibitory neurons, which were modeled without TA, naturally showed no difference in $\theta$.

To assess the difference in synaptic currents in a way that allowed direct comparison to $\theta$, we refer to the membrane voltage $V_m$, which is largely independent from and complementary to the firing threshold, and driven mostly by presynaptic inputs, resets after spikes notwithstanding. As the immediate cause of neuronal spiking, the overall pattern of $V_m$ (Fig 4C) largely reflected that seen in spiking activity. The contrast (right panel) shows not only notably higher membrane voltages in *A-as-dev* around the typical response time (note the orange band running along the response front), but also greater hyperpolarization starting immediately after due to post-spike resets.

Based on these measures, we then asked why there were more spikes in *A-as-dev* trials. We first focused on the contribution of $\theta$ alone to confirm that thresholds are lower in *A-as-dev* than *A-in-con*, and that these lower thresholds cause more firing. We quantified the contrast in $\theta$ at the start of target trials as $\Delta T_i^A = {}^{A\text{-}as\text{-}dev}T_i^A - {}^{A\text{-}in\text{-}con}T_i^A$, where $T_i = \theta_i(t=0)$, corresponding to the leftmost datapoints of the contrast histogram in Fig 4B, and correlated this with the target trial response size contrast $\Delta R_i^A$ across excitatory neurons. In the sample network (Fig 4D), we found that thresholds were clearly lower in *A-as-dev* than *A-in-con* ($Z = -20$, $p = 4.09 \times 10^{-89}$, median $= -0.165$ mV; one-sided Wilcoxon, $n = 800$ excitatory neurons), and that this was accompanied by a greater target trial response at the level of individual neurons (Pearson's $\rho = -0.394$, $p = 2.2 \times 10^{-31}$; one-sided Wald test, $n = 800$). As shown in Fig 4E, this relationship held across data sets: The median of $T_i^A$ among excitatory neurons was lower in *A-as-dev* than *A-in-con* ($\Delta T_{exc}^A < 0$: $Z = -8.16$, $p = 1.67 \times 10^{-16}$, grand median $= -0.094$ mV; one-sided Wilcoxon, $n = 120$), indicating that networks were less adapted during oddball sequences. In addition, the correlation across excitatory neurons between $\Delta T_i^A$ and $\Delta R_i^A$ was almost universally negative ($Z = -9.4$, $p = 2.68 \times 10^{-21}$, median $= -0.357$). This indicates that the lower the thresholds of neurons remained in *A-as-dev* compared to *A-in-con*, the more strongly they responded to target stimuli.

Yet, as we saw in Fig 4C, the membrane voltages also differed between the two sequences. Since firing is defined as the membrane voltage crossing the firing threshold, differences in either of these two factors could explain differences in activity. Therefore, considering instances of greater activity in *A-as-dev* than *A-in-con*, we wanted to know to what extent these differences were caused by lower thresholds, rather than higher membrane voltages. To this end, we calculated the relative positive contributions of $\theta$ and $V_m$ to instances (i.e., histogram bins) with greater activity as described in Methods. The resulting histogram is shown in Fig 4F. We see that in most bins, particularly along the response front, the major contribution was provided by $V_m$, as indicated by the blue shading, while greater activity after the main response, as well as during its very early portion, were driven largely by lower $\theta$. A similar

picture emerges when controlling for the magnitude of the activity difference and averaging across neurons, as shown in Fig 4G, and is maintained across datasets, as shown in Fig 4H. This indicates that adaptation initiated and dominated the stronger firing at the start of trials, whereas the later response was enlarged primarily as a result of greater presynaptic excitation. In short, lower thresholds kickstarted a self-reinforcing burst of activity.

Having established that the greater response to A in *A-as-dev* trials than *A-in-con* trials was driven by lower thresholds before stimulation, we next turned to the cause of these lower thresholds. A priori, since threshold adaptation is a direct reflection of activity, and since the response to target trials was larger in *A-as-dev* than in *A-in-con*, the lower thresholds in *A-as-dev* must have been caused by lower activity in non-target trials, i.e., by lower responses to B in *A-as-dev* than to B through E in *A-in-con*. Yet, since activity was modulated by thresholds as shown above, and since the *A-as-dev* sequence was random, implying that $^{A\text{-}as\text{-}dev}T^A \approx {}^{A\text{-}as\text{-}dev}T^B$, we might expect the lowered thresholds in *A-as-dev* to yield not only higher target responses, but also higher non-target responses. To resolve this apparent paradox, we will turn to an analysis of the spatial arrangement of activity and adaptation.

First, we mapped the average value of $\theta$ at the start of target trials, $T_i^A$, to the spatial location of the neurons, as shown in Fig 5A. In the *A-in-con* sequence, $^{A\text{-}in\text{-}con}T_i^A$ was roughly evenly distributed across the network, consistent with the random, distributed nature of stimulation. Conversely, in the *A-as-dev* sequence, a small number of neurons near the non-target stimulation site (B) were strongly adapted, whereas most of the network was less adapted, consistent with the data shown above.

Visualizing the likely cause of this difference, the non-target responses, in the same fashion (Fig 5B), we see that the pattern of activity in non-target trials, constituting 80% of all trials, closely matched that of the thresholds, as expected. Notably, although the comparison is

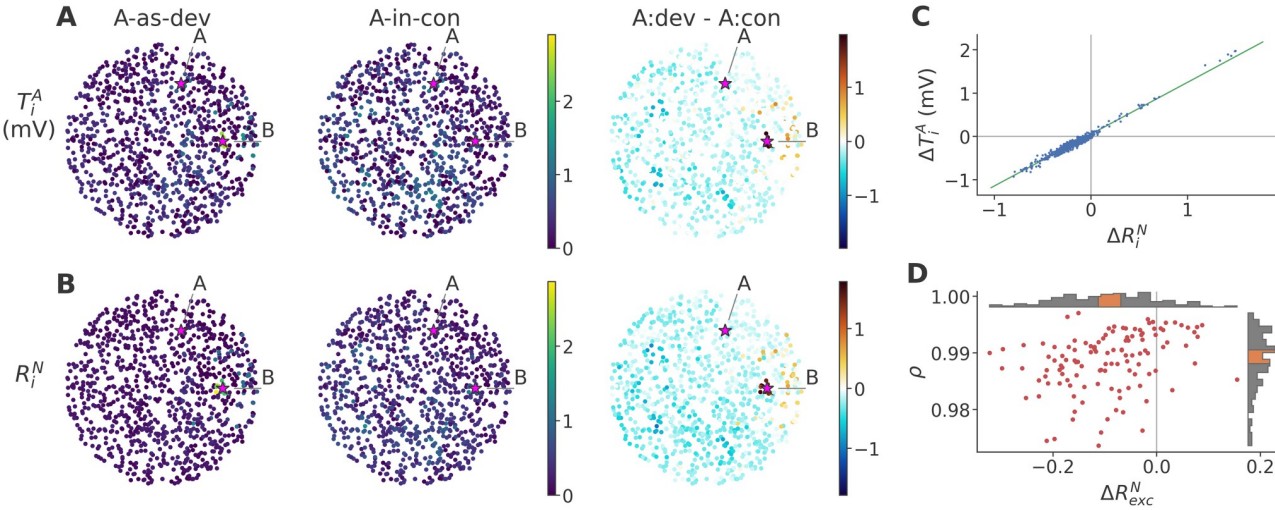

**Fig 5. Lower thresholds in *A-as-dev* are a consequence of lower *A-as-dev* non-target response magnitude. A**: $T_i^A$ in the sample network, with each neuron shown in its spatial location. Left and middle column, raw averages in the *A-as-dev* sequence ($^{A\text{-}as\text{-}dev}T_i^A$) and *A-in-con* sequence ($^{A\text{-}in\text{-}con}T_i^A$), respectively; right column, contrast between these two ($\Delta T_i^A$). A and B stimulus locations are highlighted in the left plot; see Fig 1 for the locations of the remaining stimuli used in the *A-in-con* sequence. **B**: Average response (in spikes per trial) to non-target trials ($R_i^N$, with N meaning B in *A-as-dev*, and B, C, D, and E in *A-in-con*). Columns are arranged as in panel A. **C**: Relationship across excitatory neurons of the sample network between $\Delta R_i^N$ and $\Delta T_i^A$, and associated linear regression. **D**: Pearson correlation coefficients $\rho$ of the relationship shown in the previous panel ($\Delta R_i^N$ vs $\Delta T_i^A$ across excitatory neurons), plotted against the corresponding $\Delta R_{exc}^N$ across data sets. Each marker corresponds to one network and target stimulus. Red markers indicate a significant positive correlation (one-sided Wald test, significance level 0.05). Inset histograms show the distribution of the corresponding values, with orange portions indicating the 95% confidence interval of the median, calculated by bootstrap with 10000 iterations.

between 400 B trials in *A-as-dev*, and only 100 B trials (alongside the same number of C, D and E trials) in *A-in-con*, only a small handful of neurons responded more to non-target stimulation in *A-as-dev* than in *A-in-con*, most of these very close to the stimulation site of B.

Correlating $\Delta R_i^N$ with $\Delta T_i^A$ across excitatory neurons (Fig 5C), we found that the weaker non-target response in *A-as-dev* over *A-in-con* almost perfectly predicted the resulting lower TA levels before target stimuli (Pearson's $\rho$ = 0.985). This finding held across data sets, as shown in Fig 5D, with a median correlation coefficient of 0.99 and a consistently lower median non-target response $R^N$ in *A-as-dev* than *A-in-con* sequences ($Z = -8.13$, $p = 2.12 \times 10^{-16}$, median = $-0.094$; one-sided Wilcoxon, $n = 120$), clearly validating our notion that the weaker non-target trial response was the primary driver of lower TA, which we showed above to be responsible for the higher target response in *A-as-dev* than in *A-in-con*.

Finally, we examined the cause of the lower response in non-target trials. Between the *A-in-con* sequence and the *A-as-dev* sequence, two things differ: Firstly, stimulations with non-target C, D and E in *A-in-con* are replaced with B in *A-as-dev*, which would cause different responses even in the absence of adaptation. Secondly, the resulting frequent presentation of B in *A-as-dev* changes TA values, likely causing adaptation to B and thereby decreasing the average response. We hypothesized that this adaptation is mediated by a small subset of neurons that (1) responds very quickly and rather strongly to stimulation in B (and therefore more to non-target stimuli in *A-as-dev* than *A-in-con*), and (2) has higher TA levels as a result, weakening the response of the remaining downstream neurons. Note that the logic here is equivalent to the analysis in Fig 4, where we showed that lower TA in early responders leads to a greater downstream response.

To show this, we sought to isolate this subset of early responders to B. In a given network and stimulus assignment, we ranked each neuron by the minimum response time (i.e., time to first spike) across B trials in all sequences. The ranking in the sample network is shown in Fig 6A. We found that the response developed in an approximately radial pattern, spreading outward in all directions from the stimulus site. We did notice that the early response appeared to gravitate towards the stimulus site for A, but confirmed that this was the case in all sequences separately, indicating a structural rather than dynamic cause (see S7 Fig). We then notionally split the network into an early portion, containing the first 100 neurons to respond to B, and a late portion, containing the remaining 900 neurons. In Fig 6B, we related the non-target response contrast, $\Delta R_i^N$, to the contrast in $\theta$ at the start of B trials, $\Delta T_i^B$. Across excitatory neurons, a strong linear relationship (Pearson's $\rho$ = 0.987, $n = 800$) echoed the related finding in Fig 5C, where we correlated $\Delta R_i^N$ to $\Delta T_i^A$. As the color coding and inset histograms show, however, there was a clear separation between the early and late portions of the network, with many early neurons (red, left plot) responding more strongly in the *A-as-dev* sequence, while the later portion of the network (right plot) responded more strongly in *A-in-con*. Correspondingly, the TA levels of the early portion also tended to be higher, in clear contrast to the later portion.

Across data sets, we found even stronger evidence that the early portion of the network drove adaptation to B. We first confirmed that $\Delta R_i^N$ and $\Delta T_i^B$ were robustly positively correlated (median Pearson's $\rho$ 0.992: $Z = 9.51$, $p = 9.86 \times 10^{-22}$; one-sided Wilcoxon, $n = 120$). In each network, we then identified the early and late portions with respect to stimulation in B and quantified within these two portions the median non-target response difference between *A-as-dev* and *A-in-con* ($\Delta R^N$, Fig 6C) and the median difference in $T^B$ ($\Delta T^B$, Fig 6D). We found that the response in the early portion was consistently greater in *A-as-dev* than *A-in-con* ($\Delta R_{early}^N > 0$: $Z = 8.78$, $p = 8.48 \times 10^{-19}$), unlike the late portion, which responded less ($\Delta R_{late}^N < 0$: $Z = -8.73$, $p = 1.23 \times 10^{-18}$; see also Fig 5D). $\Delta T^B$ followed the same pattern, with

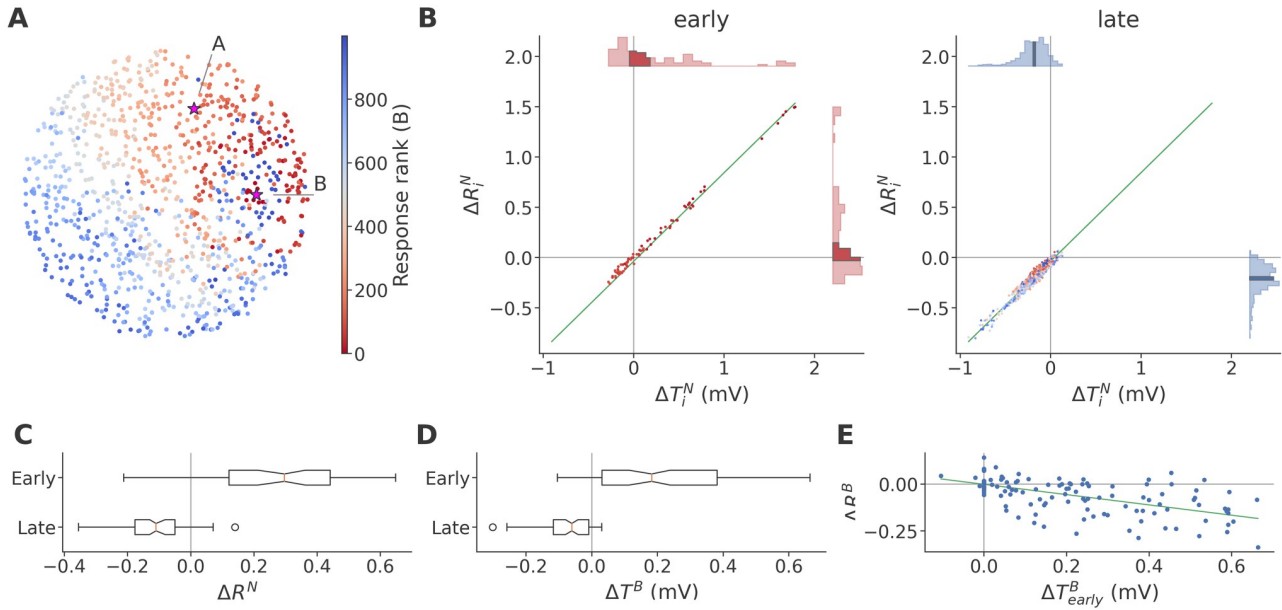

**Fig 6. Non-target responses are lower in *A-as-dev* than *A-in-con* due to adaptation in early responders to B. A**: The sample network, colored by latency rank, i.e. the rank order of the time to first spike in response to stimulation at site B in any sequence. The response develops asymmetrically due to structural properties, not due to interactions with A. Note that non-responding neurons are ranked last and appear dark blue. **B**: Relationship across excitatory neurons of the sample network between $\Delta R_i^N$ and $\Delta T_i^B$, and associated linear regression. Neurons are colored by latency rank as in panel A. Additionally, excitatory neurons among the earliest 100 responders are shown in the left plot, while later responders are shown in the right plot. Inset histograms show the distributions of the plotted values, with the 95% confidence intervals of their medians, calculated by bootstrap with 10000 iterations, in darker color. **C**: Median difference between *A-as-dev* and *A-in-con* in the non-target response $R^N$ among the first 100 neurons to respond to stimulation in B, labeled "early", and among the later neurons, labeled "late", across data sets. **D**: Median difference between *A-as-dev* and *A-in-con* in $T^B$ among the early and late neurons, respectively, across data sets. **E**: Relationship across data sets between $\Delta T_{early}^B$ and $\Delta R^B$, and associated linear regression.

the early portion exhibiting stronger TA in *A-as-dev* ($Z = 8.4$, $p = 2.21 \times 10^{-17}$), while the late portion was instead more recovered ($Z = -8.37$, $p = 2.97 \times 10^{-17}$).

Finally, to show that high $^{A\text{-}as\text{-}dev}T_{early}^B$ was causal in reducing the response to B, we correlated the median difference between *A-as-dev* and *A-in-con* in $T^B$ among the early portion, $\Delta T_{early}^B$, with the network-wide median difference of the response to B, $\Delta R^B$, as shown in Fig 6E. Across data sets, we found a strong negative correlation ($\rho = -0.587$, $p = 9.56 \times 10^{-13}$, Wald test, $n = 120$), indicating that greater TA in the neurons responding early was indeed predictive of a lowered response to B in the overall network.

To sum up, we showed in Fig 4 that lower $^{A\text{-}as\text{-}dev}T^A$ in the portion of the network responding earliest to A played an important role in increasing the overall response to A. Here, we showed that the neurons responding earliest to B exhibited higher $^{A\text{-}as\text{-}dev}T^B$, which analogously reduced the response of the network as a whole to B. We also showed that this greater $^{A\text{-}as\text{-}dev}T_{early}^B$ was a consequence of a stronger response to non-target stimulation in the early portion, or in other words, a direct consequence of the more frequent presentation of B.

Thus, we have finally traced the greater response to A in the *A-as-dev* sequence to its roots and revealed two complementary roles of TA: The frequent presentation of non-target stimulus B in the *A-as-dev* sequence and concomitant greater average activity in a subset of neurons responding early to B causes *local fatigue* by TA in this subset, leading to smaller network responses. By contrast, stimuli in the *A-in-con* sequence are presented more sporadically, allowing local fatigue to decay, thus leading to larger network responses. This difference in

non-target response sizes—small in *A-as-dev*, large in *A-in-con*—is then reflected in network-wide *global fatigue*, such that the network is more excitable to stimulation with A (or other infrequent inputs) in the *A-as-dev* sequence than in the control context.

## The role of STD in true deviance detection

As noted previously, STD alone exerted no true DD effect at a trial onset asynchrony of 500 ms, but its addition did enhance the true DD effect established by TA. To understand the basis of this apparent synergy, we contrasted the TA only condition examined in the previous section against the same networks evaluated in the full model with both TA and STD. On average, adding STD caused a reduction in response magnitude across the board, as we would expect from a depression mechanism (Fig 7A). However, this reduction was not uniform: Target responses $\overline{\boldsymbol{R}^A}$ were reduced most in the *A-as-std* sequence (more than *A-in-con*: $Z = -1.67$, $p = 0.0474$; one-sided Wilcoxon, $n = 120$), and least in the *A-as-dev* sequence (less than *A-in-con*: $Z = -2.54$, $p = 0.00554$).

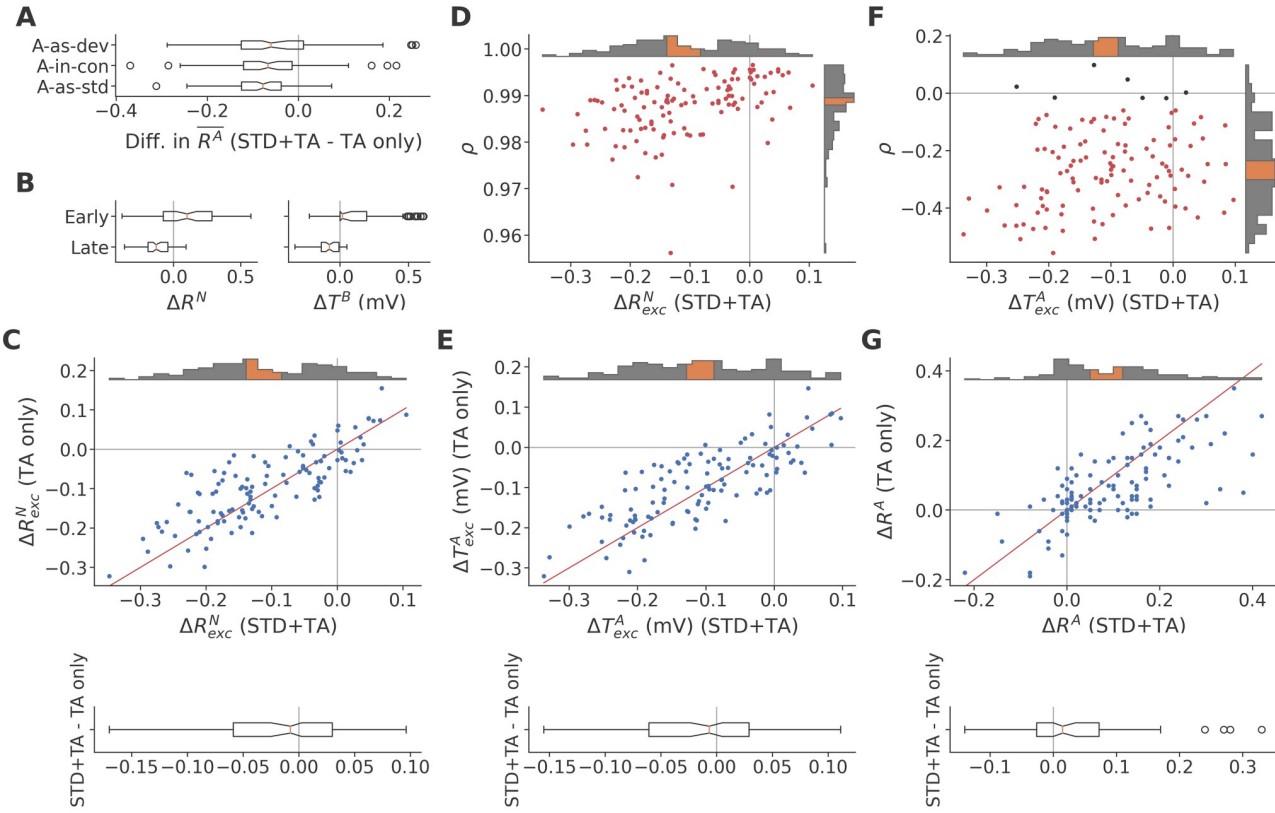

**Fig 7. Adding STD to the model slightly enhances the effects of TA. A**: Reductions in mean response size $\overline{\boldsymbol{R}^A}$ in each sequence as a consequence of adding STD to the model, across data sets. **B**: $\Delta\boldsymbol{R}^N$ (top) and $\Delta\boldsymbol{T}^B$ (bottom) among early and late neurons, respectively, across data sets, in the full model with both STD and TA. Compare to Fig 6C and 6D. **C**: $\Delta\boldsymbol{R}^N_{exc}$ across data sets in the full model (horizontal axis) and the ablated model with TA only (vertical axis). The red line is the identity function; points above this line represent networks where the difference between responses in *A-as-dev* and *A-in-con* is greater in the full model than in the ablated model. The subtraction of the two quantities in the box plot below shows that the difference in non-target responses tends to be greater when STD is included in the model. **D**: Pearson correlation coefficients $\rho$ of the correlation between $\Delta\boldsymbol{R}^N_i$ and $\Delta\boldsymbol{T}^A_i$ across excitatory neurons in the full model, plotted against the corresponding $\Delta\boldsymbol{R}^N_{exc}$. Compare to Fig 5D. **E**: $\Delta\boldsymbol{T}^A_{exc}$ across data sets in the full and ablated models, arranged as in panel C. The subtraction of the two quantities in the box plot below shows that $\Delta\boldsymbol{T}^A$ tends to be more negative in the full model. **F**: Pearson correlation coefficients $\rho$ of the correlation between $\Delta\boldsymbol{T}^A_i$ and $\Delta\boldsymbol{R}^A_i$ across excitatory neurons in the full model, plotted against the corresponding $\Delta\boldsymbol{T}^A_{exc}$. Compare to Fig 4E. **G**: $\Delta\boldsymbol{R}^A_{exc}$ across data sets in the full and ablated models, arranged as in panel C. The subtraction of the two quantities in the box plot below shows that the target response difference between *A-as-dev* and *A-in-con* tends to be greater in the full model.

To understand how STD affects true DD and why it reduces *A-as-std* responses more than *A-as-dev*, we first examined the effects of TA in the full model, following the analysis on the ablated model laid out in the previous section. Starting at the cause of the weaker response in non-target trials, we show in Fig 7B the full model equivalent of Fig 6C and 6D. The differences in non-target response magnitude between *A-as-dev* and *A-in-con*, $\Delta \boldsymbol{R}^N$, and in the resulting $\boldsymbol{T}^B$ levels both retained a clear separation between the portion of neurons responding early, which showed greater responses ($Z = 4.75$, $p = 1.02 \times 10^{-6}$; one-sided Wilcoxon, $n = 120$) and higher TA levels ($Z = 5.21$, $p = 9.51 \times 10^{-8}$), and the neurons responding later, whose response ($Z = -8.77$, $p = 9.19 \times 10^{-19}$) and TA levels ($Z = -8.39$, $p = 2.49 \times 10^{-17}$) were lower in the *A-as-dev* sequence than in *A-in-con*. Likewise, the correlation across excitatory neurons between these two contrasts remained positive ($Z = 9.51$, $p = 9.86 \times 10^{-22}$) and very strong (median Pearson's $\rho = 0.992$). This indicates that local fatigue to B mediated by TA in the early responders remained a key driver of the weaker response to stimulation in non-target trials.

Comparing the non-target response differences in Fig 7C, we found that they were on average larger in the full model than in the ablated model ($Z = 2.11$, $p = 0.0175$; one-sided Wilcoxon, $n = 120$), with the grand medians of $\Delta \boldsymbol{R}^N_{exc}$ 36% greater in the full model than in TA only. Based on the ablated model analysis in Fig 5, we expected the even weaker non-target responses in *A-as-dev* compared to *A-in-con* to coincide with even lower TA levels at the start of A trials. Indeed, $\Delta \boldsymbol{R}^N_i$ remained tightly correlated with $\Delta \boldsymbol{T}^A_i$ (median Pearson's $\rho = 0.989$), as shown in Fig 7D. Correspondingly, $^{A\text{-}as\text{-}dev}\boldsymbol{T}^A_{exc}$ was lower than $^{A\text{-}in\text{-}con}\boldsymbol{T}^A_{exc}$, as shown by way of their difference in Fig 7E, and this too by a slightly greater amount in the full model than in the ablated model ($Z = 1.95$, $p = 0.0257$).

Finally, based on the ablated model analysis in Fig 4, we sought to confirm that lower $^{A\text{-}as\text{-}dev}\boldsymbol{T}^A$ was responsible for the stronger target response. As shown in Fig 7F, $\Delta \boldsymbol{T}^A_{exc}$ and $\Delta \boldsymbol{R}^A_{exc}$ were negatively correlated ($Z = -9.43$, $p = 2.09 \times 10^{-21}$, median $= -0.268$; one-sided Wilcoxon, $n = 120$), though the correlation was weaker in the full model ($Z = -5.13$, $p = 1.49 \times 10^{-7}$). Despite the weaker correlation, however, the full model target response remained clearly stronger in *A-as-dev* than *A-in-con*, as shown in Fig 7G, and this too more so in the full model than in the ablated model ($Z = -3.19$, $p = 7.23 \times 10^{-4}$), with the grand medians of $\Delta \boldsymbol{R}^A$ 94% greater in the full model than in TA only. This greater difference, of course, directly reflects the greater DDI reported in Fig 3.

Taken together, the data indicate that, with STD added to the model, the overall chain of causality—greater local fatigue to B in *A-as-dev* than *A-in-con* resulting in weaker B responses and less global fatigue, which in turn allows for stronger target responses—remained intact with only minor quantitative changes. The primary cause for the greater target response, therefore, appeared to be the smaller non-target response (Fig 7C), which was where we noted the first clear shift towards increased true DD.

This naturally raises the question of why, with the addition of STD, the non-target *A-as-dev* response was reduced more than non-target *A-in-con* responses. We hypothesized that, due to its relatively short time constant, the direct effect of STD across trials was limited to repeated trials of the same stimulus, such as the frequent stimulus in oddball sequences. This would mean that, in our model, STD is capable only of local fatigue, reducing the non-target response in *A-as-dev*, but not of global fatigue, leaving networks with only STD unable to differentiate between *A-as-dev* and *A-in-con* target trials.

We had good reasons for this assumption: Updates to the spike-triggered depression variable $x_j$ (cf. Eq (4)) would decay to a negligible $e^{\frac{-1s}{\tau_x}} = 0.0013$ times the original value within two trials, making it unlikely that inputs separated by intervening trials could be "remembered" in

the network. Conversely, the frequent presentation of stimulus B could likely cause enough depression to suppress its own response.

To show in the sample network that the model does not maintain a history beyond the most recent trial, we correlated $^{A-as-dev}R_j^A$ with $x_j$ at the start of subsequent trials across excitatory neurons. We found that, while the correlation was strong and very clearly positive in the trial immediately following target (Pearson's $\rho = -0.983$, $p = 0$; Wald test, $n = 800$), it was almost completely lost by the next trial ($\rho = -0.0642$, $p = 0.0696$) under the combined influence of recovery and unrelated activity in the intervening trial. Similar behavior was also seen in the *A-in-con* sequence, as shown in S5 Fig.

To show that STD being carried over between repeated trials was responsible for the greater reduction of non-target responses in *A-as-dev* than in *A-in-con*, however, we needed to estimate the extent to which activity of a given neuron was affected by changes in STD. Unlike $\theta_j$, which directly affects the activity of neuron $j$, $x_j$ only exerts its effects postsynaptically. Therefore, we devised a measure $\boldsymbol{D}$ as detailed in Quantifying the effect of STD, which estimates the postsynaptic impact of STD in terms of membrane potential lost.

As noted above, we hypothesized that STD is directly responsible only for a reduction of non-target *A-as-dev* trials, relative to *A-in-con*, whereas differences between target *A-as-dev* and *A-in-con* trials are not a result of STD directly. To test this, we calculated $\boldsymbol{D}$ for A and B trials and contrasted it across *A-as-dev* and *A-in-con* sequences as with $\boldsymbol{T}$ before. As shown in Fig 8A, we find that the response to A (top) reached the entire network in both sequences, but clearly experienced greater depression in *A-in-con* than in *A-as-dev* across most of the network. In contrast, the response to B (bottom) in the *A-as-dev* sequence was limited to a small

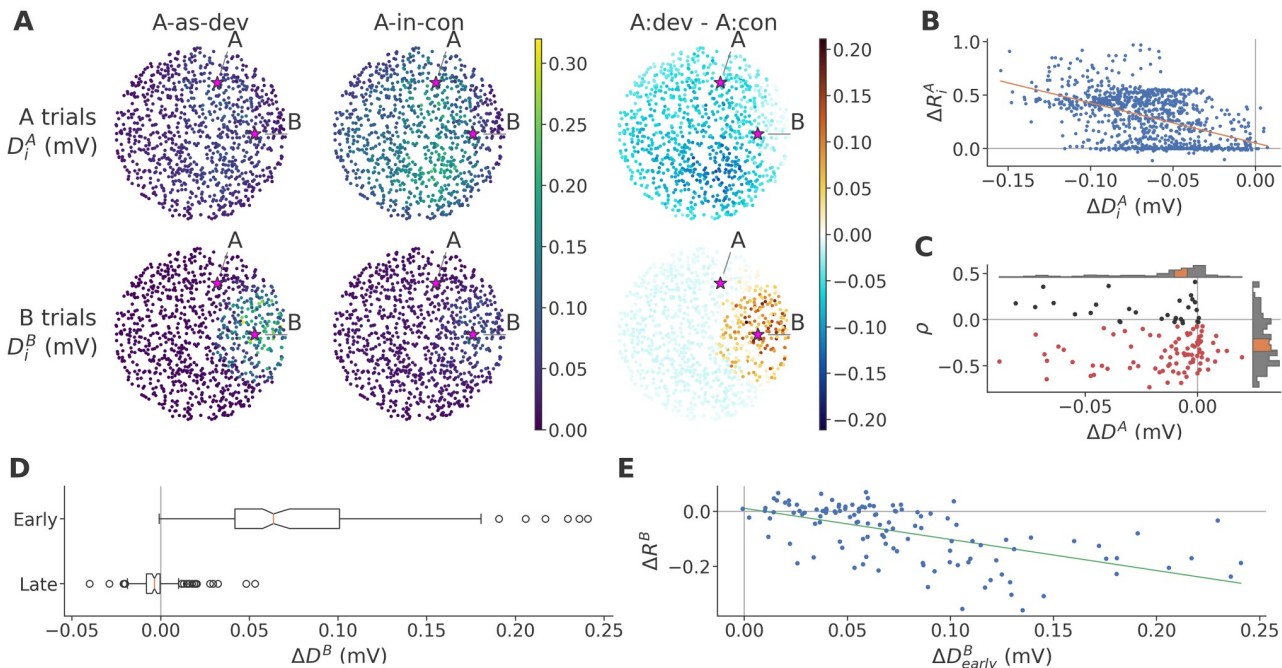

**Fig 8. In the full model, STD reduces *A-as-dev* non-target and increases *A-as-dev* target responses relative to *A-in-con*. A**: Average depression $\boldsymbol{D}_i$ in target (A) trials (top row) and B trials (bottom row) in the *A-as-dev* (left) and *A-in-con* sequence (center), and their contrast (right) in the sample network. The color scale is shared across columns (and between *A-as-dev* and *A-in-con*) for direct comparability. **B**: Relationship across neurons of the sample network between $\Delta \boldsymbol{D}_i^A$ and $\Delta \boldsymbol{R}_i^A$, and associated linear regression. **C**: Pearson correlation coefficients $\rho$ of the correlation between $\Delta \boldsymbol{D}_i^A$ and $\Delta \boldsymbol{R}_i^A$, plotted against the corresponding $\Delta \boldsymbol{D}^A$ across data sets. **D**: $\Delta \boldsymbol{D}^B$ in the first 100 neurons to respond to B, and in the remaining, late portion of the network, across data sets. **E**: Relationship across data sets between $\Delta \boldsymbol{D}_{early}^B$ and $\Delta \boldsymbol{R}^B$, and associated linear regression.

patch near the stimulated area, which consequently experienced stronger depression than in the *A-in-con* sequence. As a result, the contrast $\Delta D_i^B$ was slightly negative in the periphery, but strikingly elevated near the stimulation site, indicating much greater synaptic depression in *A-as-dev* than *A-in-con*.

To show that $D$ captures not only changes in $V_m$, by definition, but also in the resulting response, we correlated $\Delta D_i^A$ with $\Delta R_i^A$ across neurons of the sample network. Notice that this mirrors the correlation shown in Fig 4 between $\Delta T_i^A$ and $\Delta R_i^A$. As shown in Fig 8B, the relationship was clearly negative (Pearson's $\rho = -0.496$, $p = 2 \times 10^{-63}$; Wald test, $n = 1000$), indicating that higher $D$ causes lower responses as expected. We confirmed this relationship across data sets, see Fig 8C, and found that greater depression in *A-as-dev* predicted a lower response in most networks ($Z = -7.61$, $p = 1.37 \times 10^{-14}$; one-sided Wilcoxon, $n = 120$). We note that calculating this same correlation in the model with STD only yields no significantly negative dependence ($Z = 0.945$, $p = 0.828$; see S4 Fig).

We believe that the failure of higher $D$ to predict lower responses (in some networks with the full model, and in most networks with STD only) is due in part to the very low strength of depression, with differences between sequences even smaller, and in part to the way $D$ is calculated, which does not exclude silent neurons, nor differentiate between depressed inputs arriving before or after postsynaptic activity. In the latter case, given that most neurons fire at most one spike per trial, the depression is ineffective, but nonetheless counts towards $D$.

Having gained a rough intuition of how individual neurons are affected by STD, we now turn to its effect on the network, working with statistics across data sets. First, in Fig 8D, we recapitulate the logic established in Fig 6: Given the spatial clustering of $D_i^B$, we surmised a similar local fatigue effect as with TA. We found that, indeed, presynaptic input to neurons responding early to stimulation in B was more depressed in *A-as-dev* than *A-in-con* ($\Delta D_{early}^B > 0$: $Z = 9.5$, $p = 1.01 \times 10^{-21}$; one-sided Wilcoxon, $n = 120$), while the late portion was subject to slightly less depression ($\Delta D_{late}^B < 0$: $Z = -3.65$, $p = 1.33 \times 10^{-4}$). We can relate this directly to the non-target response differences in the early and late portions shown in Fig 7A, which followed a similar pattern, and take this to indicate that the more frequent presentation of B in *A-as-dev* causes local synaptic depression.

To show that greater local fatigue by STD reduces the non-target response, we correlated $\Delta D_{early}^B$ with $\Delta R^B$, echoing Fig 6E. As we show in Fig 8E, higher $\Delta D_{early}^B$ was clearly predictive of a lower network response ($\rho = -0.581$, $p = 1.81 \times 10^{-12}$; Wald test, $n = 120$). This confirms that STD-mediated local fatigue played a role in weakening the non-target response in the *A-as-dev* sequence.

Finally, given the very clear difference between *A-as-dev* and *A-in-con* in $\Delta D_i^A$ in the sample network, we considered the possibility that, in the context of the full model, STD could also evoke global fatigue in the *A-in-con* sequence, contrary to our stated hypothesis. If so, we should expect to see less $D^A$ in *A-as-dev* than *A-in-con*. To confirm this, we analyzed $\Delta D^A$ across data sets and found it to be robustly negative ($Z = -7.52$, $p = 2.66 \times 10^{-14}$; one-sided Wilcoxon, $n = 120$), as can be seen in the inset histogram in Fig 8C. We conclude that there is clear evidence for greater STD-mediated global fatigue in *A-in-con* than *A-as-dev* in the full model.

To sum up, in the full model, STD acted in much the same way as TA, causing both a reduction in *A-as-dev* of the response to the frequently presented non-target stimuli (lower $^{A\text{-}as\text{-}dev}R^B$) by local fatigue, and an increase in the response to target stimuli (higher $^{A\text{-}as\text{-}dev}R^A$) by relief from global fatigue, relative to *A-in-con*. Yet, as we showed in Fig 7 and in the previous section, a weaker $^{A\text{-}as\text{-}dev}R^B$ also entailed lower thresholds $^{A\text{-}as\text{-}dev}T^A$, which independently led to higher target responses $^{A\text{-}as\text{-}dev}R^A$. Thus, adding STD to the model increased the DDI both directly,

through its intrinsic effects in local and global fatigue, and indirectly, by altering the pattern of TA by way of local fatigue of the response to B.

We note that the direct and indirect effects of STD on global fatigue appear to be on the same order of magnitude. To see this, compare the change in $\Delta T^A$ as a result of adding STD to the model (Fig 7E, bottom) with the corresponding value of $\Delta D^A$ (Fig 8E), both of which are on the order of $\approx 0.1$ mV. Conversely, notice that, since TA and STD work in similar fashion to increase the *A-as-dev* target response relative to *A-in-con*, it is very plausible that TA likewise increases the effect of STD, by enhancing the STD-mediated local fatigue of B and thereby increasing the difference in STD-mediated global fatigue. Thus, this synergy allows the individually weak STD to exert a significantly larger effect when paired with the stronger TA.

## Spatial basis of local fatigue

Finally, we sought to confirm whether "local" and "global" fatigue, which we defined so far by the latency of response, did in fact reflect a spatial property of the network. To do so, we assessed the distance between the center of the target stimulation site and each neuron in all data sets, and calculated $R^A$, $T^A$ and $D^A$ as a function of this distance. Fig 9A–9C shows that

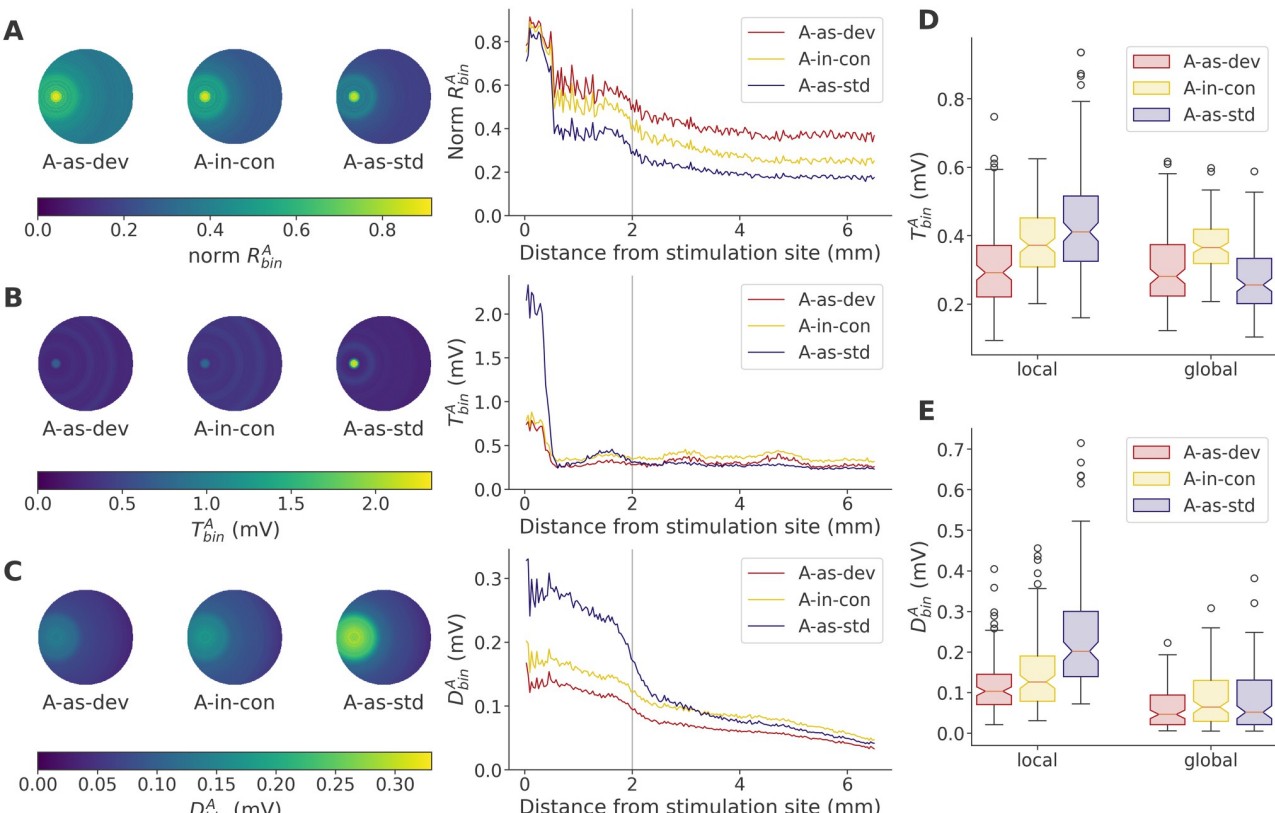

**Fig 9. Fatigue acts in spatially distinguishable subpopulations. A**: Response size $R^A$ as a function of distance from the center of the stimulation site, evaluated across all data sets in 200 bins between 0 mm and 6.5 mm, and normalized by the response size in recovered networks (initial trial). Each of the 200 data points represents the mean response across all relevant trials of all neurons in the corresponding distance bin. Left, spatial illustration, using all data with a fictive stimulus location. Right, the same data plotted as curves. The vertical line indicates the chosen local/global divider. **B**: As in panel A, but showing $T^A$ without normalization. **C**: As in panel A, but showing $D^A$ without normalization. **D**: Mean $T^A$ in neurons $< 2$ mm from the stimulation site (local) and more distant neurons (global). Data are averaged across relevant neurons and trials in each dataset, providing 120 data points to be summarized by box plot. Comparisons of interest are between *A-as-std* and *A-as-dev* in the local subset, and between *A-in-con* and *A-as-dev* in the global subset. **E**: Corresponding mean $D^A$ in local and global portions of the network.

there were indeed clear spatially defined differences between *A-as-dev*, *A-in-con* and *A-as-std*. Responses, normalized by the response to stimulation in the recovered networks (Fig 9A), clearly spread out farthest in *A-as-dev*, and least in *A-as-std*, despite a nearly indistinguishable response at the immediate stimulation site. For thresholds, $^{A-as-std}T^A$ (Fig 9B) was much stronger near the stimulation site, and somewhat weaker distally, than either of the less frequent paradigms. A similar pattern emerged with $D^A$ (Fig 9C), though here, the local patch covered a larger area, as we might expect from a synaptic effect.

To quantify this statistically, we split the neurons into local and distal (global) subsets, cutting at a radius of 2 mm from the stimulus site, yielding an average local subset size of $226.8 \pm 13.2$ neurons, out of 1000. Note that while this overlaps with the "early" subset used above, some early neurons are non-local, and some local neurons respond late or not at all (see S6 Fig). We then assessed the difference of the mean values of $T^A$ and $D^A$ across the two subsets in each sequence, as shown in Fig 9D and 9E. In keeping with the previously stated analysis, we found that local fatigue was higher in *A-as-std* than in *A-as-dev* both for $T^A$ ($Z = -8.24$, $p = 8.31 \times 10^{-17}$; one-sided Wilcoxon, $n = 120$) and for $D^A$ ($Z = -7.58$, $p = 1.71 \times 10^{-14}$), indicating that the network response to the non-target stimulus in the *A-as-dev* sequence was indeed kept in check by high local fatigue. Similarly, we found that global fatigue was greater in *A-in-con* than in *A-as-dev*, again for both $T^A$ ($Z = -6.66$, $p = 1.37 \times 10^{-11}$) and $D^A$ ($Z = -7.08$, $p = 7.27 \times 10^{-13}$), indicating that the *A-in-con* sequence produced more fatigue across the wider network, leading to smaller target responses than in the *A-as-dev* sequence. Incidentally, this greater global fatigue, caused by the greater diversity of stimuli presented in *A-in-con* than in *A-as-dev*, was also reflected in greater local fatigue around the target stimulus site for $T^A$ ($Z = -7.33$, $p = 1.15 \times 10^{-13}$) and $D^A$ ($Z = -5.7$, $p = 6.13 \times 10^{-9}$), providing an additional pathway for a comparatively large response in *A-as-dev*.

As a side note, the apparent wave pattern in TA (Fig 9B) likely has two causes. The distal local maxima are only seen in TA and likely reflect high TA levels at the neighboring stimulation sites, located 2.9 mm and 4.8 mm away. The local maximum around 1.5 mm, by contrast, also appears in unnormalized $R$ (see S8 Fig) and likely reflects an early response in the band between 1 mm and 2 mm from the stimulus site, which is reached by excitatory stimulated neurons but not inhibitory due to the connectivity structure. This band and the chosen distance threshold for the local patch coincide not by conscious choice, yet may raise doubts about the validity of our analysis. However, the gradual decline in $R$ and $D$ beyond the threshold suggests that the principle remains sound, and true DD in our model can indeed by described well by a distance dependence of fatigue.

## Influence of stimulation paradigm

We tested two ways in which the stimulus paradigm might influence the expression of true DD. Firstly, since our data contained both nearby stimulus pairs (2.9 mm apart) and distant pairs (4.8 mm apart), it is reasonable to ask whether the nearby pairs were more affected by local fatigue. As shown in Fig 10A, these two groups did not differ in their average deviance detection index ($t[118] = 0.71$, $p = 0.479$, independent samples t-test), indicating that greater proximity between stimulus locations in the oddball sequences did not degrade the true DD capability in general. We note that, unlike e.g. in auditory cortex, where a particular stimulus evokes responses only in a small part of the network, the situation here is that a majority of neurons responds to any given stimulus, that is, the receptive fields of the neurons overlap to a very large degree, which may explain the absence of a distance dependence.

We did find, however, that the DDI varied more among nearby stimulus pairs ($F[2,60] = 9.92$, $p = 0.00208$, Brown-Forsythe test for equality of variances). Without investigating this in

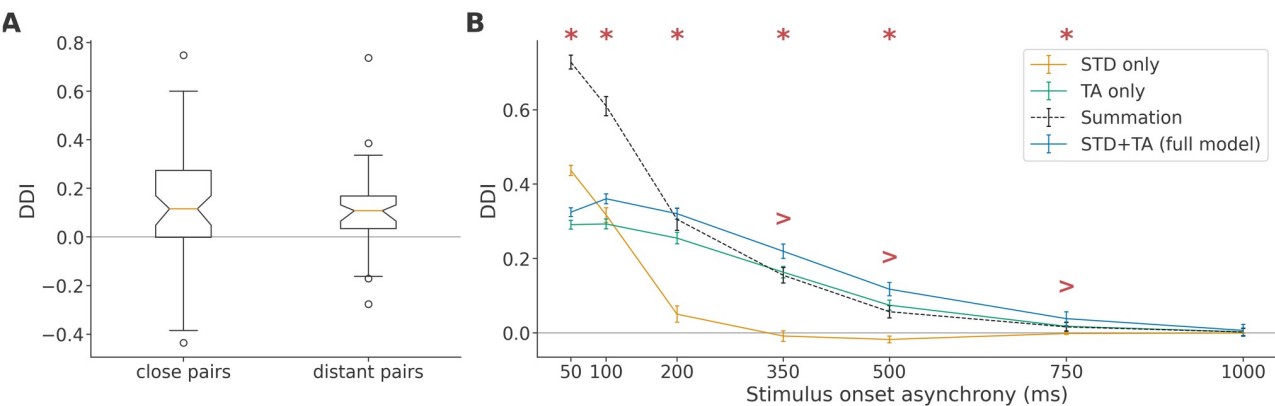

**Fig 10. Stimulation time scale, but not location, affects true DD. A**: Deviance detection index of close and distant pairs of stimulus locations ($n = 60$ each). **B**: Deviance detection indices (mean +- SEM across data sets) as a function of stimulus onset asynchrony. Plotted are the ablated models with only one plasticity mechanism, their summation (dashed), and the full model DDI. Asterisks show significantly positive DDI for the full model, established with a one-tailed Wilcoxon signed-rank test at a significance level of 0.05. Greater-than markers show significant differences between summation and full model, established in the same manner.

detail, we suspect that while the greater proximity may have led to an overlap in early-responding populations, the overlap may have been sparse enough to yield both smaller deviant responses in some cases, through fatigue of excitatory pathways, and larger deviant responses in others, through fatigue of isolated inhibitory pathways. We further note that the cell culture data in [39], Fig 4, show a similar trend towards greater variability in overlapping stimulus pairs.

Secondly, we investigated the impact of stimulus onset asynchrony (SOA). The apparent inability of STD to induce a significant DDI (see Fig 3) stands in contrast to the preceding literature, where STD, not TA, was used successfully to model deviance detection [32–34]. Given that the stimulus onset asynchrony (SOA, 500 ms) was much greater than the STD time constant (150 ms), but shorter than the TA time constant (1 s), it seemed likely that this discrepancy was involved in exaggerating the effect of TA relative to STD in our analysis. To confirm this, we systematically manipulated the SOA and quantified the DDI of the model with either one or both of the plasticity mechanisms.

As shown in Fig 10B, we find that STD is entirely capable of inducing true DD by itself at an SOA of 200 ms and below. By contrast, the DDI of the TA-only model becomes moderately larger with shorter onset asynchronies, but shows signs of saturation and is surpassed by the STD-derived DDI at an SOA of 50 ms. Neither mechanism induces any significant DDI at a long SOA of 1 s. In addition, we find that the synergy effect, where the DDI of the full model significantly exceeds the sum of the DDIs of the two mechanisms individually, extended over onset asynchronies of 350ms, 500 ms and 750 ms. However, at shorter SOAs, we instead find that the full model DDI was smaller than the sum of the ablated model DDIs, which we tentatively interpret as due to saturation.

## Discussion

### Deviance detection emerges from encoding regularities

In this study, we showed that non-synaptic neuronal adaptation can induce true deviance detection in a randomly wired, but spatially organized network of simple leaky integrate-and-fire neurons. Neuronal adaptation achieves this, as illustrated in Fig 11, by suppressing the response of neurons local to the frequently stimulated site. This drastically reduces the response to frequent stimulations at an early stage of signal propagation through the network,

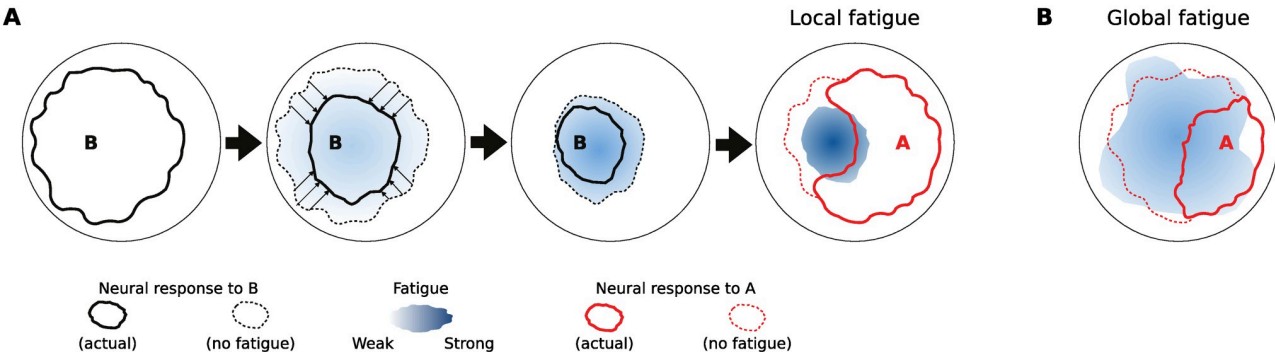

**Fig 11. Illustration of the origin of DD in our model. A**: The situation in *A-as-dev*. Frequent presentation of non-target B causes strong local fatigue around the stimulation site for B, preventing the response from spreading to the wider network. Therefore, when target stimulus A is presented (right), large parts of the network are recovered, which responds vigorously and widely as a consequence. **B**: The situation in *A-in-con*. Stimuli rarely repeat, allowing local fatigue around stimulus sites to recover, and thus responses to spread widely, causing fatigue throughout the network. Therefore, when target stimulus A is presented, the network is globally fatigued and responds less vigorously than in *A-as-dev*.

allowing the more distant parts of the network to recover from adaptation and respond vigorously to infrequent deviant stimuli. By contrast, a more varied paradigm including inputs at multiple stimulation sites allows local fatigue to partly recover between stimulus presentations, resulting in a greater response through greater recruitment of the wider network. As a result, the network becomes more fatigued, diminishing the response to each individual stimulus. True deviance detection, then, is a side effect of encoding and adapting to the regularity of the frequently presented stimulus, which leaves the wider network less adapted and more responsive than usual.

## Biological basis of neuronal threshold adaptation

Adaptation of the neuronal firing threshold to past activity or membrane depolarization has been demonstrated at a wide variety of timescales ranging from milliseconds [36, 55, 56], supporting precise spike timing [57, 58], to days [59], supporting activity homeostasis. History dependence on a timescale of hundreds of milliseconds to several seconds, as used in our work, has been described in various systems, including early sensory areas [60, 61] and later processing stages up to primary sensory cortices and beyond [35, 36, 50, 62, 63]. Commonly described as spike-frequency adaptation (SFA), this intrinsic feature of neurons supports contrast adaptation [35, 64], enhances the signal-to-noise ratio [65, 66], and improves signal coding properties of the wider network [67–69]. SFA can be induced by a number of mutually independent mechanisms, including a buildup of inactivation in voltage-gated sodium channels [48, 70], and increased gating of calcium- and sodium-dependent potassium channels [47, 49, 64, 71, 72] and calcium-gated chloride channels [73, 74]. Perhaps due to this variety in underlying mechanisms, how to model SFA in integrate-and-fire neurons has been debated, and models have been proposed using both hyperpolarizing currents [75, 76] and adaptive thresholds [77–79]. While there are subtle differences between current- and threshold-based models, and theoretical work has shown current-based approaches to be somewhat superior in capturing firing rate dynamics [80, 81], recent large-scale model fitting studies have done very well with adaptive threshold models similar to that used here [46, 82].

## Threshold adaptation supports true deviance detection

It has been suggested that non-synaptic adaptation such as TA lacks stimulus specificity and is therefore unable to support true DD [34, 83]. In addition, a frequent argument made is that

the various timescales involved in stimulus-specific adaptation (SSA) in an oddball context [84] is best mirrored in the similar range of synaptic plasticity timescales [42, 85]. Yet, in the auditory system, where DD has been studied most intensely, there is intrinsic adaptation at all stages, from the auditory nerve [86, 87] through subcortical stations [88] to auditory cortex [63]. The timescales involved exhibit a hierarchy, going from tens to hundreds of milliseconds in the sensory and subcortical neurons to several seconds in cortex, a range likewise very similar to that seen in oddball SSA [84]. Furthermore, this hierarchy is accompanied by a similar hierarchy in SSA, with thalamus and inferior colliculus being sensitive to shorter stimulus intervals than cortex [89]. Finally, it has been shown that combining intrinsic adaptation with STD imbues networks with exquisite sensitivity to changes in the input signal [90]. Having shown that TA can support true DD in an oddball context, we anticipate that future models of DD and its analogues will consider including TA or other forms of intrinsic plasticity in addition to synaptic depression.

## Comparison to other models of DD

In many ways, our model closely resembles the more elaborate and explicitly cortical model proposed by [34], which uses short-term synaptic depression as its adaptation mechanism. Local fatigue is implemented in their model in synaptic depletion in the cortical column sensitive to the target stimulus and is greatest for the standard and least for the deviant, with the many-standards paradigm consistently evoking significant activity across the wider network and inducing an elevated background fatigue. In addition, both their model and earlier experimental results [91] show occasional large responses to standard, which we have also observed anecdotally. Notably, however, our model operates without explicit compartmentalization into columns. Furthermore, [34] report a fine balance of parameter tuning, outside of which their model ceases to operate as intended and does not detect deviant inputs.

Here, we did not explore parameter sensitivity beyond the dependence on stimulus frequency. However, several reasons lead us to believe that our model is relatively robust to parameter choices. Firstly, we did not have to search parameter space to find true DD, but readily found clear evidence for it. Secondly, the spatial organization of our network model is simple and continuous, such that spatial scaling is likely to affect the results quantitatively rather than qualitatively. Thirdly, our structural constraints are mild (e.g., no dependence of connection probability, weight, etc. on postsynaptic neuron type), resulting in a model of low structural complexity and therefore, we surmise, low complexity in its qualitative characteristics. Finally, the dynamic complexity of our network is likewise low, as is evident from the propagation of responses through the network (cf. Fig 4), making it unlikely that our results derive from intricate interactions between model components.

Given the mild structural constraint in our model noted above, we speculate that our interpretation of the source of true DD is applicable to a wider range of neural architectures, including non-cortical circuitry. On the flipside, the divergence in our model between a heavily recurrent structure on the one hand and a largely one-way response pattern on the other hand makes it difficult to judge whether DD arises as a result of lateral inhibition or adaptation of lateral excitation as in [30, 34, 92], as the model structure might indicate, or, conversely, as a result of adaptation in feed-forward connections as in [32], as the response dynamics might suggest. We speculate that the robustness of our model derives at least in part from this ambiguity: Like in the model by [33], TA reduces all activity propagation, regardless of its destination, thus reducing the impact of structural properties.

In contrast to the predominant thinking in the field (see e.g. [24] for an extensive review), which suggests that active cancellation may be required for true DD, our work suggests that both SSA and true DD can be mediated by a single mechanism, and that true DD in particular does not require complex connectivity, as noted above. Indeed, while stronger responses to stimuli in deviant over control contexts have been reported in cortex of many mammals, including humans [20], other primates [93], and rats [31, 53, 94], recent work has shown true DD even in dissociated neuronal culture [39], where connectivity is assumed to be comparatively simple, much like in our model. Rather than invalidating work dissecting the circuit-level mechanisms of cortical DD [95–97], our model may provide a possible explanation for the demonstrated presence of true DD in subcortical sensory areas [27] and suggests that what Parras and colleagues identify as prediction errors may in fact persist in the absence of top-down predictions.

Finally, a comparison between models is incomplete without mention of the magnitude of true DD quantified in the DDI. Across our datasets, DDI at an SOA of 500 ms varied substantially, with the bulk of values contained between 0 and 0.2. This is similar to experimental data from cell culture probed at the same SOA [39]. Likewise, [34] report in their model values around 0.2–0.3 for pairs of nearby stimulus sites, while at greater separations their DDI is well above 0.5. Of the model variants in [32], the ABC variant, which is most similar to ours and was fitted to experimental data [98], shows substantially larger values around 0.4, though also at only half the deviant occurrence rate.

## Adaptation may support subcortical DD

There is indeed some evidence for this. If feedback signals within cortex and from cortex to midbrain sensory areas carry predictions that cancel out incoming activation, as is commonly assumed in the predictive coding framework [3, 5, 99], then we should expect these signals to interfere with adaptation-dependent processing, since neurons silenced by top-down predictions are unable to build up a stimulus history with adaptation or depression. Thus, knocking out predictive feedback should facilitate such adaptation-driven DD, while any DD based on predictive inputs would disappear. Indeed, such differential effects have been reported by [100]. Under the influence of NMDA blockers, which inhibit long-term plasticity in cortex and therefore prevent the formation of accurate predictive models, late components of DD in an oddball paradigm are reduced, while components within the first 10 ms to 20 ms after stimulus onset are enhanced. This suggests that early DD, likely driven by subcortical processing [27, 38], may be driven by adaptive mechanisms such as that demonstrated in our work, whereas later, cortical DD derives largely from more sophisticated computations involving explicit prediction and error signals [101, 102]. It would be very interesting to explore this interaction in a hierarchical model that combines the adaptation-based approach in early, subcortical areas with a predictive approach [29, 30, 103] in higher, cortical areas.

## TA and STD synergize to yield supralinear scaling

Our second main finding is that synaptic short-term depression, which acts in similar fashion to neuronal adaptation both in terms of how it is induced and in how it affects activity flow in the network, enhances the DD effect of neuronal adaptation, and vice versa. We demonstrated that this synergy is a result of increased local fatigue of the response to the frequently presented stimulus by STD, which decreases global fatigue by both mechanisms in oddball sequences and allows the deviant response to appear larger compared to control. Conversely, without strong local fatigue in control sequences, global fatigue is greater, leading to smaller individual responses.

This synergy can be understood as a simple nonlinear effect of adaptation. While the effect on local fatigue of adding STD is direct and mediated only by STD, the resulting reduction of global fatigue in the oddball sequence depends entirely on the pre-existing fatigue levels mediated by TA and is indeed counteracted to some extent by STD. We speculate that increasing the strength of TA, rather than adding STD, would have a similar effect. Thus, prior to saturation, each additional unit of adaptation from any sufficiently similar source would lead to increasing gains in DD. While it is a priori unsurprising that short-term plasticity should give rise to nonlinearities, the direction of the nonlinearity typically reflects the direction of plasticity: Presynaptic facilitation leads to supralinear activation of the post-synaptic side, and vice versa, depression leads to sublinear activation [42, 85], which is typically reflected also in population-level activity [104]. The reverse has been shown in the context of inhibition-stabilized networks [105] as a result of the so-called paradoxical effect that increasing excitatory drive decreases activity [106, 107]. However, we have no reason to believe that our model is inhibition-stabilized and are not aware of other work showing supralinear scaling from depression. It would be interesting to confirm our intuition in the model presented here, as well as in a context of more realistic neural and network dynamics.

In addition, while interactions between different forms of short-term plasticity have been investigated in detail at the level of synaptic processes [42, 85, 108–111], we are aware of only two studies on the interaction between synaptic and neuronal short-term plasticity, investigating oscillations [112] and spike train irregularities [113] caused by such interactions, respectively. A principled investigation of the interactions between short-term plasticity mechanisms is missing and would seem to be an important addition to our understanding of neural circuit dynamics.

## Differences between TA and STD

In the present work, we chose parameters such that TA had a much stronger effect than STD, particularly with the chosen time scales. We believe that the chosen time constants fall within a physiologically plausible range: For synaptic depression, 150 ms, chosen based on thalamocortical inputs to primary sensory areas [43, 44], is at the lower end of the plausible range for cortical neurons, with values reported ranging from hundreds of milliseconds to several seconds for intracortical synapses [42, 85, 114–116]. For threshold adaptation, time constants have been reported in a similar range, from hundred of milliseconds [36, 50, 63, 82] to several seconds [47–50]. The stark contrast in apparent capabilities between STD and TA in our model, therefore, is likely a result of the choice of time constants. In particular, with the higher time constants for STD reported intracortically, we expect STD to likewise contribute significantly to true DD at longer stimulus intervals.

In one aspect, however, neuronal and synaptic plasticity clearly differ: While TA throttles the output of the affected neurons, STD throttles the input to downstream targets. A re-examination of Fig 9B and 9C, for example, shows that STD affects more neurons as a direct result. The greater reach of the immediate effect of STD perhaps explains why TA has often been overlooked for deviance detection, besides the mistaken claim of stimulus specificity. We also note that this greater reach may result in greater overlap between the neurons experiencing fatigue from several stimulus classes. This, in turn, would mean that STD may facilitate the kinds of interactions between different stimuli that give rise to DD, suggesting its applicability to a wider range of neural architectures and of stimuli than TA. For example, in primary auditory cortex, where neurons are tuned along a tonotopic gradient, and are located near and connected to similarly tuned neighbors, STD would be better placed than TA to cause a dependence of DD on the pitch separation between presented tones.

## Impact of spontaneous activity

In comparison to biological sensory processing systems, our model is, of course, very simple. One key limitation is in the activity dynamics: Unlike living systems, our model does not produce any spontaneous activity, and evoked responses die out quickly. In contrast, even dissociated cell cultures show some amount of ongoing activity in absence of stimulation. Greater activity would likely substantially affect our results by preferentially following less adapted pathways and therefore reducing the contrast between adapted and recovered parts of the network. Greater spontaneous activity has indeed been linked to reduced DD in rodent inferior colliculus [83, 117] and auditory cortex [118], and [63] showed that increased background activity also attenuates the level of adaptation to stimuli in individual neurons.

Relatedly, the lack of spontaneous activity also implies that no explicit predictive signals could possibly be generated in our model. Such signals would be necessary for the network to respond to the omission of expected stimuli, as has been demonstrated in cortex of humans [119, 120], marmosets [121], and rats [122]. However, we are unaware of any work showing conclusively the presence or absence of omission responses in subcortical areas. Evidence from EEG presented by [119] suggests that no such responses may occur, but a finer-grained analysis in animal models would help to further narrow down to what extent the early, subcortical DD processes are driven by prediction and adaptation [101]. Along with other considerations outlined above, the absence of evidence of omission responses seems to suggest that our conceptual model of local and global fatigue may be a valid description of subcortical processing.

## Early and late response in culture

In dissociated culture [39], the stimulus response is biphasic, with a peak of directly evoked activity in the first 10 ms, and a longer response lasting up to 200 ms. In the results reported by Kubota and colleagues (Fig 2), true DD appears to become evident only in the later response. Since we do not simulate conduction delays, the response in our model is compressed over a much shorter time period. However, we also see a larger response difference in the later stage of the response, as activity passes from the stimulated neurons and their immediate neighborhood into the wider network. It remains to be confirmed whether the distinction between early and late responses in culture is tied to separate neuron populations, as in our model, indicating some amount of linearity of activity propagation, or whether the early and late responses both develop in a distributed manner throughout the network.

## Limitations and future work

In addition to the limitations noted above, such as the lack of intrinsic activity, our work is limited in a number of other ways. Firstly, in constructing our model, we followed some rather outdated modeling practices, particularly in relation to inhibitory neurons and their synapses, which we modeled without adaptive processes. However, GABAergic neurons are known to be capable of both cellular adaptation and short-term synaptic plasticity [123–125]. In addition, even our excitatory synapses only depress, while of course facilitating and depressing-facilitating synapses are likewise common in cortex [42, 85]. It is difficult to predict how these additional plasticity mechanisms would affect our results. We may speculate at obvious immediate effects—adaptation in inhibitory neurons and facilitation in excitatory synapses would likely work to counteract the processes detailed in our work—but, as we have shown, interactions between plasticity mechanisms can be surprising and complex.

Further, across both inhibitory and excitatory neurons, many simplifying assumptions were made for expedience, many of which (e.g. cellular, synaptic, and structural homogeneity)

bear closer examination and better grounding in physiology in future work. Even within the confines of point-process LIF models, much greater biological plausibility could be achieved. Similarly, the manner of stimulus delivery, consisting of an instantaneous, unadaptive excitation that overcomes most or all adaptation in the stimulated neurons, is faithful to the equivalent situation in cell culture, but clearly not representative of e.g. thalamic input to primary sensory cortex. Subsequent investigations targeting particular brain areas and sensory domains should aim to incorporate plausible, if not realistic, neuron models, connectivity, dynamics, and input characteristics in order to better understand the roles of TA and STD in the mismatch responses observed in the experimental literature.

## Conclusion

As alluded to above, we anticipate that future models of DD and its attendant signatures (SSA, mismatch negativity, etc.) will rely on hierarchical structure, including not just different cortical areas [33, 92], but also subcortical areas with their own processing dynamics. Given the evidence presented here, we strongly believe that neuronal adaptation will play a key role in such future models, working alongside synaptic short-term plasticity [34] and long-term plasticity [30] to better explain the full richness of temporal sensory processing.

## Supporting information

**S1 Fig. The equivalent of Fig 4 in the full model with both TA and STD.**
(PDF)

**S2 Fig. The equivalent of Fig 5 in the full model with both TA and STD.**
(PDF)

**S3 Fig. The equivalent of Fig 6 in the full model with both TA and STD.**
(PDF)

**S4 Fig. The equivalent of Fig 8 in the ablated model with only STD.**
(PDF)

**S5 Fig. The STD variable $x_i$ only correlates with the activity of the immediately preceding trial.** Each marker represents the coefficient $\rho$ of the Pearson correlation between the spike counts in a target trial and the corresponding values $x_i$ at the beginning of the immediately following trial ($\Delta$ trials = 1) or after one intervening trial ($\Delta$ trials = 2). Recall that $x$ is reduced upon spiking, which implies that an effective correlation is negative. Color represents the p-value of the alternative hypothesis that $\rho < 0$. Horizontal lines indicate the median $\rho$ and p. Horizontal jitter is introduced to aid visual separability of the markers.
(PDF)

**S6 Fig. Early and local populations overlap to some degree.** Distance to stimulus site A plotted against the response rank (see Fig 6) in the sample network. Vertical and horizontal grey lines represent the cutoffs for the "early" subset (Fig 6) and the "local" subset (Fig 9), respectively. Notice that, while many neurons are both early and local, or both late and global, a substantial number of neurons are early and global, or late and local.
(PDF)

**S7 Fig. Response latency to stimulus B in the sample network, separated by sequence.** See Fig 6A.
(PDF)

**S8 Fig. Response size, TA and STD as a function of distance from the stimulation site. A**: Unnormalized response size *R*, mean across all datasets. The three sequence averages, which show a pronounced local maximum around 1 mm to 2 mm, are shown in addition to the recovered response (i.e., the response to stimulation in A outside of sequence context), which was used for normalization (B, Fig 9A). **B-D**: As in Fig 9A–9C.
(PDF)

## Author Contributions

**Conceptualization:** Felix Benjamin Kern, Zenas C. Chao.

**Data curation:** Felix Benjamin Kern.

**Formal analysis:** Felix Benjamin Kern.

**Funding acquisition:** Zenas C. Chao.

**Investigation:** Felix Benjamin Kern.

**Methodology:** Felix Benjamin Kern, Zenas C. Chao.

**Project administration:** Zenas C. Chao.

**Resources:** Zenas C. Chao.

**Software:** Felix Benjamin Kern.

**Supervision:** Zenas C. Chao.

**Validation:** Felix Benjamin Kern.

**Visualization:** Felix Benjamin Kern.

**Writing – original draft:** Felix Benjamin Kern.

**Writing – review & editing:** Felix Benjamin Kern, Zenas C. Chao.

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
