## [Decision Letter · Decision Letter 0]

26 Jul 2023

Dear Dr Kern,

Thank you very much for submitting your manuscript "Short-term neuronal and synaptic plasticity act in synergy for deviance detection in spiking networks" for consideration at PLOS Computational Biology. As with all papers reviewed by the journal, your manuscript was reviewed by members of the editorial board and by several independent reviewers. The reviewers appreciated the attention to an important topic. Based on the reviews, we are likely to accept this manuscript for publication, providing that you modify the manuscript according to the review recommendations.

Sincerely,

Jonathan David Touboul

Academic Editor

PLOS Computational Biology

Thomas Serre

Section Editor

PLOS Computational Biology

Reviewer's Responses to Questions

**Comments to the Authors:**

Reviewer #1: the authors use two synaptic mechanisms -- fast short term depression (STD) and slow threshold adaptation (TA) -- to explain deviance detection (DD). The consideration of TA is indeed novel and the supralinear effect of STD and TA is an interesting result worth publishing. I have major concerns which will require substantial rewriting. However, I believe these changes are straightforward to address and will significantly improve the paper.

Primary comments:

-Kern and Chao appear to use DD and stimulus-specific adaptation (SSA) interchangeably throughout the paper, when they should not. SSA is considered a neural correlate of DD but very much distinct from DD. See https://doi.org/10.1007/s00422-014-0585-7 (which I see is a paper the authors have referenced but perhaps have not read carefully).

It appears that stimulus-specific adaptation (SSA) would be appropriate for Kern and Chao to use because the ability of the network to generalize its deviance detection outside of the oddball stimulus is not tested or considered. Thus, DD should be replaced with SSA throughout the entire paper. One exception would be where a reference specifically mentions DD. If a paper mentions DD, it should be made separate from discussions about SSA. Similarly, if a paper only mentions SSA, it should not be included in a discussion about DD. One example includes a reference on line 590 for Abolafia et al 2011., where the paper explicitly mentions SSA and not DD.

-line 56: Studies of DD must carefully distinguish between repetition suppression and true deviance detection (Ross and Hamm, 2020). While this is true, the reference clearly mentions that SSA and DD are distinct and some ways they can overlap in particular experiments. The remainder of this paragraph goes on to distinguish DD and repetition suppression and correctly uses the definition of DD, but I am not sure if the authors understand that SSA is not simply repetition suppression. DD is defined according to statistics because the stimuli in DD can be far less regular than the oddball stimulus (the oddball stimulus has a repeated, regular train of tones with an occasional oddball tone that still follows the same time interval. In contrast, true DD would be able to adapt to irregular but statistically uniform sounds and detect an occasional tone that does not fit the statistics of the "usual" sounds.

-Another important note is that mismatch negativity (MMN), DD, and SSA, are distinct phenomena. Kern and Chao should again be careful about references that refer to MMN, DD, and/or SSA. I leave it to the authors to clearly establish the difference between these three phenomena and rewrite the introduction. From here on I will assume the authors use SSA in place of DD.

-Line 64: While many existing computational models of SSA do indeed place a significant focus on synaptic plasticity as the causal mechanism, an extremely important mechanism involves lateral traveling waves. This is a mechanism that Yarden and Nelken (2017) heavily rely on to generate SSA (in addition to population spikes, but the population spikes are closely related to the traveling wave). Indeed, Kern and Chao's network possibly contains a wave-like mechanism as noted on line 238, where there are neurons that respond earlier and later as the stimulus response propagates through the network.

-At a glance, the remainder of the paper appears to be sound, with obvious great care put into the design of the model and metrics. I will look more closely at the results once the authors address my primary concerns above, but my impression is that the paper is quite strong.

Secondary comments:

-Could the authors please comment on the choice of circular domain? If it is justified somewhere in the paper, it would be good to include it in the paragraph on line 113.

Reviewer #2: In this study, the authors demonstrated the important role of neuronal adaptation (implemented by threshold adaptation, TA) in supporting true deviance detection (DD), which was usually ignored or rejected in other modeling studies (e.g., Mill et al., 2011; Yarden and Nelken, 2017). This main message is well elucidated through meticulous thinking characterized by careful consideration of all relevant factors. The author also brought up the countereffect on the adaptation mechanism by inhibition (either via lateral inhibition or top-down prediction) in the discussion, which I think is worth mentioning, as the direct relationship between adaptation-based and prediction-based mechanisms is not often addressed in the studies of DD. This work would benefit future studies that consider the possibility of various forms of plasticity in sensory processing. I support the eventual publication of this manuscript in Plos Computational Biology. Most of my comments focus on clarity and readability.

Comments:

(-----clarity-----)

(1)The naming of {’dev’, ‘std’} and {’target’, ‘non-target’} is a bit confusing in the first place (even though it is described in lines129-136) because usually ‘dev’ means the ‘target’ rather than a condition. A re-consideration of the naming may improve the clarity.

(2)The mechanism of TA in DD is well described (in Figures 4, 5, and 6) and is convincing. This may also account for a related experimental observation, where the amplitude of a deviance response depends on the distance between the ‘dev’ and ‘std’ tones: A larger distance elicits a larger deviance response. This phenomenon is reproduced by an adaptation-based model (e.g., Fig. 2B in May and Tiitinen, 2010). The authors mentioned using different pairs of sites for A/B tones (lines 137-138). Could the authors comment on whether their model reproduces such a phenomenon?

(3)Line 141: I am not certain what the 4 data sets are. Could the authors make it clearer?

(4)Line 197: The authors mention the second peak (at around 20 ms) in the std condition. The second peak only happens in the STD+TA condition. This seems to be not observed in reality. Could the authors comment on this?

(5)The authors use a simple model (without too many constraints and fine-tuning of parameters) so that the underlying mechanisms of DD can be elucidated more clearly. It would be good to add more explanation about the simplification and the corresponding effect. For example, (a) the decision to present stimuli instantaneously rather than over a period (e.g., 50 ms), (b) the decision to arrange the five sites circularly rather than linearly.

(6)The result in this work shows no significant DDI that differs from 0 in the STD-only condition (line 222). Is it due to the instantaneous stimuli or other configurations (e.g., time constant, SOA) that differ from other modeling works emphasizing STD?

(7)In line 426, 0.0013 is calculated using 1s in the numerator, why not 500ms (the SOA)?

(8)Line 611: A DDI map of two factors (e.g., \\tau_{STD} and \\tau_{TA} within a reasonable range) may help with the robustness in the parameter space.

(9)In line 621, providing a brief description of the predominant thinking will help readability.

(-----Accuracy, Scientific formality-----)

(10)In Equation 1, a resistance variable R on the right-hand side is missing. Usually, it should be something like [V_{rest}-V_m] +R*I_{syn} so that they are in the same unit. Need some clarifying text if only a variable I_{syn} is used here.

(11)In line 93, the expression “with = 30 ms the membrane time constant and = −60 mV the resting membrane potential” can be written as “with the membrane time constant = 30 ms and the resting membrane potential = −60 mV”. The rest of the contents have the same problem.

(12)Line 155: “...with A and B in the msc sequence and calculated z-scores with respect to the population” is not clear to me. Maybe some equations would help?

(13)Line 189: Measurements such as equation 6 can be described in the Method section (and maybe also equations 7 and 8)

(14)Lines 221-225: For comparisons of STD with 0, the authors employed a two-tailed Wilcoxon test, considering the possibility of either an increase or decrease in values. Conversely, for comparisons of TA with 0, a one-tailed Wilcoxon test was used. Please provide the reason(s); otherwise, the statistical method (one-tailed or two-tailed) should remain consistent. However, the authors did not explicitly state the comparison method when assessing the full model against 0. This ambiguity exists throughout the whole paper. I recommend providing the name of the statistical method used for this comparison to ensure consistency in statistical analyses.

(15)The statistical parameters (e.g., p-value p) should be italicized.

(16)The Wilcoxon statistic results are typically stated in terms of Z-score rather than T-score.

(-----Figures-----)

(17)Figure 1: The colors for Exc and Inh in Fig.1A and Fig. 1B are not consistent. The color contrast for sites 1-5 in Fig.1C is too low, which makes it hard to identify them in the network. Marking sites 1-5 directly on the graph may help with clarity.

(18)Figure 3: Panel C is mentioned in the legend but missing in the figure.

(19)Figure 4 Legend: Description in F, G, and H is difficult to understand.

(20)Figure 5: Can add A and B circles on all subplots for clarity.

(-----Proofreading-----)

(21)Lines 139-140: (labeled B to E) or (labeled “B” to “E”)? need to be consistent here.

(22)Figure 2 legend: “DDI, see Equation eq:ddi” A link to the equation is missing.

(23)Line 153: 120 networks or 120 data sets?

(24)Line 221: Figure 3C or Figure 3B?

(25)Line 275: “during oddball sequences”, or more specifically “during dev sequences”?

(26)Proofreading may be needed. I assume I didn’t find all of them.

Reviewer #3: Review is uploaded as an attachment

**Have the authors made all data and (if applicable) computational code underlying the findings in their manuscript fully available?**

Reviewer #1: Yes

Reviewer #2: Yes

Reviewer #3: Yes

PLOS authors have the option to publish the peer review history of their article (what does this mean?). If published, this will include your full peer review and any attached files.

Reviewer #1: No

Reviewer #2: No

Reviewer #3: **Yes: **Tohar S. Yarden

Figure Files:

Data Requirements:

Reproducibility:

References:

---

## [Decision Letter · Decision Letter 1]

29 Sep 2023

Dear Dr Kern,

We are pleased to inform you that your manuscript 'Short-term neuronal and synaptic plasticity act in synergy for deviance detection in spiking networks' has been provisionally accepted for publication in PLOS Computational Biology.

Best regards,

Jonathan David Touboul

Academic Editor

PLOS Computational Biology

Thomas Serre

Section Editor

PLOS Computational Biology

Reviewer's Responses to Questions

**Comments to the Authors:**

Reviewer #1: Thank you for the thorough reply. It looks like I was deeply mistaken. I am sorry for the misunderstanding and the unnecessarily harsh tone.

I did not emphasize enough on my initial review that this work is very strong and easily surpasses the bar for publication at PLOS computational biology.

I have no further comments. Thank you for your contribution.

Reviewer #2: The authors have considered and incorporated my feedback and suggestions into their work. I appreciate their responsiveness to my comments, and I am confident that the final version will be a valuable contribution to the field of DD modeling.

**Have the authors made all data and (if applicable) computational code underlying the findings in their manuscript fully available?**

Reviewer #1: Yes

Reviewer #2: Yes

PLOS authors have the option to publish the peer review history of their article (what does this mean?). If published, this will include your full peer review and any attached files.

Reviewer #1: No

Reviewer #2: **Yes: **Shih-Cheng Chien

---

## [Editor Report · Acceptance letter]

8 Oct 2023

PCOMPBIOL-D-23-00635R1 

Short-term neuronal and synaptic plasticity act in synergy for deviance detection in spiking networks

Dear Dr Kern,

I am pleased to inform you that your manuscript has been formally accepted for publication in PLOS Computational Biology. Your manuscript is now with our production department and you will be notified of the publication date in due course.

With kind regards,

Zsofi Zombor
